# Spatial and temporal pattern of structure–function coupling of human brain connectome with development

**Guozheng Feng[1,2,3], Yiwen Wang[1,2,3], Weijie Huang[1,2,3], Haojie Chen[1,2,3], Jian Cheng[4], Ni Shu[1,2,3]***

[1]State Key Laboratory of Cognitive Neuroscience and Learning & IDG/McGovern Institute for Brain Research, Beijing Normal University, Beijing, China; [2]BABRI Centre, Beijing Normal University, Beijing, China; [3]Beijing Key Laboratory of Brain Imaging and Connectomics, Beijing Normal University, Beijing, China; [4]School of Computer Science and Engineering, Beihang University, Beijing, China

*For correspondence:
nshu@bnu.edu.cn

Competing interest: The authors declare that no competing interests exist.

**Abstract** Brain structural circuitry shapes a richly patterned functional synchronization, supporting for complex cognitive and behavioural abilities. However, how coupling of structural connectome (SC) and functional connectome (FC) develops and its relationships with cognitive functions and transcriptomic architecture remain unclear. We used multimodal magnetic resonance imaging data from 439 participants aged 5.7–21.9 years to predict functional connectivity by incorporating intracortical and extracortical structural connectivity, characterizing SC–FC coupling. Our findings revealed that SC–FC coupling was strongest in the visual and somatomotor networks, consistent with evolutionary expansion, myelin content, and functional principal gradient. As development progressed, SC–FC coupling exhibited heterogeneous alterations dominated by an increase in cortical regions, broadly distributed across the somatomotor, frontoparietal, dorsal attention, and default mode networks. Moreover, we discovered that SC–FC coupling significantly predicted individual variability in general intelligence, mainly influencing frontoparietal and default mode networks. Finally, our results demonstrated that the heterogeneous development of SC–FC coupling is positively associated with genes in oligodendrocyte-related pathways and negatively associated with astrocyte-related genes. This study offers insight into the maturational principles of SC–FC coupling in typical development.

## eLife assessment

This study presents a **useful** exploration of the complex relationship between structure and function in the developing human brain using a large-scale imaging dataset from the Human Connectome Project in Development and gene expression profiles from the Allen Brain Atlas. The evidence supporting the claims of the authors is **solid**, although the inclusion of more systematic analyses of structural and functional connectivity with respect to myelin measures and oligodendrocyte-related genes, and also more details regarding the imaging analyses, cognitive scores, and design and validation strategies, would have strengthened the paper. The work will be of interest to developmental biologists and neuroscientists seeking to elucidate structure-function relationships in the human brain.

## Introduction

In neural circuitry, long-range (extracortical) interconnections among local (intracortical) microcircuits shape and constrain the large-scale functional organization of neural activity across the cortex (*Vázquez-Rodríguez et al., 2019*; *Sarwar et al., 2021*; *Demirtaş et al., 2019*; *Deco et al., 2011*; *Breakspear, 2017*). The coupling of structural connectome (SC) and functional connectome (FC) varies greatly across different cortical regions reflecting anatomical and functional hierarchies (*Vázquez-Rodríguez et al., 2019*; *Valk et al., 2022*; *Zamani Esfahlani et al., 2022*; *Gu et al., 2021*; *Baum et al., 2020*) and is regulated in part by genes (*Valk et al., 2022*; *Gu et al., 2021*), as well as its individual differences relates to cognitive function (*Gu et al., 2021*; *Baum et al., 2020*). Despite its fundamental importance, our understanding of the changes in SC–FC coupling with development is currently limited. Specifically, the alterations in SC–FC coupling during development, its association with cognitive functions, and the underlying spatial transcriptomic mechanisms remain largely unknown.

Network modelling of the brain enables the characterization of complex information interactions at a system level and provides natural correspondences between structure and function in the cortex (*Zamani Esfahlani et al., 2022*; *Bassett and Sporns, 2017*). Advances in diffusion MRI (dMRI) and tractography techniques have allowed the *in vivo* mapping of the white matter (WM) connectome (WMC), which depicts extracortical excitatory projections between regions (*Feng et al., 2022*). The T1- to T2-weighted (T1w/T2w) ratio of MRI has been proposed as a means of quantifying microstructure profile covariance (MPC), which reflects a simplified recapitulation in cellular changes across intracortical laminar structure (*Valk et al., 2022*; *Paquola et al., 2019b*; *Liu et al., 2022*; *Paquola and Hong, 2023*; *Park et al., 2022*). Resting state functional MRI (rs-fMRI) can be used to derive the FC, which captures the synchronization of neural activity (*Honey et al., 2009*). A variety of statistical (*Valk et al., 2022*; *Gu et al., 2021*; *Baum et al., 2020*), communication (*Vázquez-Rodríguez et al., 2019*; *Zamani Esfahlani et al., 2022*), and biophysical (*Breakspear, 2017*; *Sanz-Leon et al., 2015*) models have been proposed to study the SC–FC coupling. The communication model is particularly useful because it not only depicts indirect information transmission but also takes into account biodynamic information within acceptable computational complexity (*Zamani Esfahlani et al., 2022*; *Avena-Koenigsberger et al., 2017*). However, most studies have relied on WMC-derived extracortical communications as SC to predict FC, while ignoring the intracortical microcircuits, the MPC. In the present study, we propose that incorporating both intracortical and extracortical SC provides a more comprehensive perspective for characterizing the development of SC–FC coupling.

Previous studies in adults have revealed that the SC–FC coupling is strongest in sensory cortex regions and weakest in association cortex regions, following the general functional and cytoarchitectonic hierarchies of cortical organization (*Vázquez-Rodríguez et al., 2019*). This organization may occur due to structural constraints, wherein cortical areas with lower myelination and weaker WM connectivity tend to have more dynamic and complex functional connectivity (*Vázquez-Rodríguez et al., 2019*; *Gu et al., 2021*). Large-scale association networks emerged over evolution by breaking away from the rigid developmental programming found in lower-order sensory systems (*Buckner and Krienen, 2013*), facilitating regional and individual specialization (*Preti and Van De Ville, 2019*). In terms of developmental changes in SC–FC coupling, a statistical model-based study (*Baum et al., 2020*) identified positive age-related changes in some regions, while fewer regions exhibited negative changes. Furthermore, there is evidence that SC–FC coupling is linked to cognitive functions in healthy children (*Chan et al., 2022*), adults (*Gu et al., 2021*; *Medaglia et al., 2018*), and patients (*Kuceyeski et al., 2019*), suggesting that it may be a critical brain indicator that encodes individual cognitive differences. Nonetheless, a more comprehensive investigation is needed to understand the precise pattern of SC–FC coupling over development and its association with cognitive functions.

Cortical SC–FC coupling is highly heritable (*Gu et al., 2021*) and related to heritable connectivity profiles (*Valk et al., 2022*), suggesting that the development of coupling may be genetically regulated. The Allen Human Brain Atlas (AHBA) (*Hawrylycz et al., 2012*) is a valuable resource for identifying genes that co-vary with brain imaging phenotypes and for exploring potential functional pathways and cellular processes via enrichment analyses (*Whitaker et al., 2016*; *Arnatkeviciute et al., 2021*; *Fornito et al., 2019*). For instance, a myeloarchitectural study showed that enhanced myelin thickness in mid-to-deeper layers is specifically associated with the gene expression of oligodendrocytes (*Paquola et al., 2019a*). Another functional study found that the expression levels of genes involved

**Table 1.** Predictive significance of the communication model.

| Predictor | | $p_{spin}$ | Predictor | | $p_{spin}$ |
|---|---|---|---|---|---|
| | Gamma values = 0.12 | 0.93 | | Weight-to-cost transformations = 0.12 | 0.84 |
| | Gamma values = 0.25 | 0.69 | | Weight-to-cost transformations = 0.25 | 0.97 |
| | Gamma values = 0.5 | 0.63 | | Weight-to-cost transformations = 0.5 | 0.90 |
| | Gamma values = 1 | 0.89 | | Weight-to-cost transformations = 1 | 0.75 |
| | Gamma values = 2 | 0.77 | | Weight-to-cost transformations = 2 | 0.90 |
| Shortest path length | Gamma values = 4 | 0.45 | Path transitivity | Weight-to-cost transformations = 4 | 0.61 |
| **Communicability** | | **<0.001** | Matching index | | 0.42 |
| Cosine similarity | | 0.25 | Greedy navigation | | 0.99 |
| | Weight-to-cost transformations = 0.12 | 0.63 | **Mean first-passage times of random walkers** | | **0.01** |
| | Weight-to-cost transformations = 0.25 | 0.59 | | **Timescales = 1** | **<0.001** |
| | Weight-to-cost transformations = 0.5 | 0.32 | | Timescales = 2.5 | 0.26 |
| | Weight-to-cost transformations = 1 | 0.72 | | Timescales = 5 | 0.91 |
| | Weight-to-cost transformations = 2 | 0.60 | Flow graphs | Timescales = 10 | 0.80 |
| Search information | Weight-to-cost transformations = 4 | 0.75 | | | |

Note: $p_{spin}$: spin test. The communication models in bold provide the optimal combination.

in calcium ion-regulated exocytosis and synaptic transmission are associated with the development of a differentiation gradient (*Xia et al., 2022*). However, the transcriptomic architecture underlying the development of SC–FC coupling remains largely unknown.

In this study, we analysed data obtained from the Lifespan Human Connectome Project Development (HCP-D) (*Somerville et al., 2018*), which enrolled healthy participants ranging in age from 5.7 to 21.9 years. Our main objective was to investigate the SC–FC coupling of brain connectome and characterize its developmental landscapes. Specifically, we aimed to determine whether the SC–FC coupling encodes individual differences in cognition during development. Finally, we explored the genetic and cellular mechanisms underlying the development of SC–FC coupling of brain connectome. To assess the reproducibility of our findings, sensitivity and replication analyses were performed with different parcellation templates, different tractography strategies, and a split-half independent validation method.

## Results

We selected 439 participants (5.7–21.9 years of age, 207 males) in the HCP-D dataset who met our inclusion criteria: available high-quality T1/T2, dMRI, and rs-fMRI data that met the quality control thresholds. For each participant, we generated multiple connectomes using 210 cortical regions from the Human Brainnetome Atlas (BNA) (*Fan et al., 2016*), which comprised MPC, WMC, and FC. Intra-cortical connectivity was represented by MPC. According to the WMC, 27 weighted communication models (*Zamani Esfahlani et al., 2022*) were calculated to characterize geometric, topological, or dynamic connectivity properties. After analysis, we found that communicability (*Crofts and Higham, 2009*), mean first-passage times of random walkers (*Noh and Rieger, 2004*), and flow graphs (timescales = 1) provided the optimal combination of extracortical connectivity properties because of significantly predicting FC ($p < 0.05$, 1000 spin test permutations, *Table 1*). We used these three models to represent the extracortical connectivity properties in subsequent discovery and reproducibility analyses (*Figure 1—figure supplement 1*).

### Spatial pattern of cortical SC–FC coupling

We used SCs (MPC and three WMC communication models) to predict FC per node based on a multilinear model (*Vázquez-Rodríguez et al., 2019*; *Figure 1*), and quantified the nodewise SC–FC

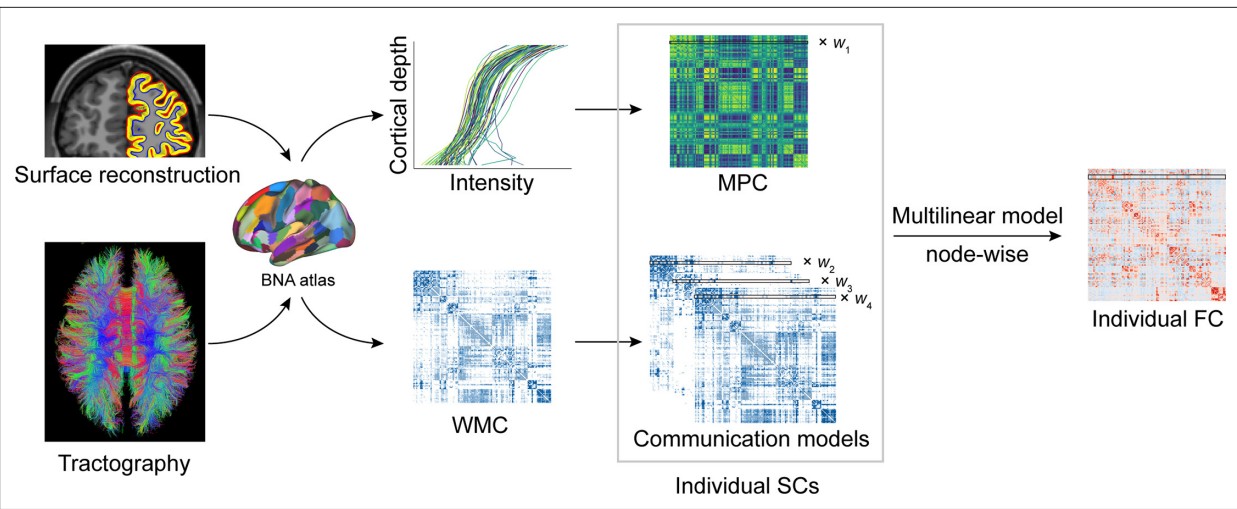

**Figure 1.** Structural connectome–functional connectome (SC–FC) coupling framework. The framework used to quantify nodal SC–FC coupling in the human brain. The microstructure profile covariance (MPC) was used to map similarity networks of intracortical microstructure (voxel intensity sampled in different cortical depth) for each cortical node. The white matter connectome (WMC) represents the extracortical excitatory projection structure, and communication models were then constructed to represent the complex process of communication. A multilinear model was constructed to examine the association of individual nodewise SC (MPC and communication models) profiles with FC profiles.

The online version of this article includes the following figure supplement(s) for figure 1:

**Figure supplement 1.** Pipeline of communication model selection and reproducibility analyses.

coupling as an adjusted coefficient of determination $r^2$. We observed that the grouped SC–FC coupling varied across cortical regions (mean adjusted $r^2 = 0.14 \pm 0.08$, adjusted $r^2$ range = [0.03, 0.45], *Figure 2A*), and regions with significant coupling were located in the middle frontal gyrus, precentral gyrus, paracentral lobule, superior temporal gyrus, superior parietal lobule, postcentral gyrus, cingulate gyrus, and occipital lobe (p < 0.05, 1000 spin test permutations, *Figure 2B*). Similar heterogeneous patterns of coupling were observed when categorizing cortical regions into seven functional subnetworks (*Yeo et al., 2011*) (visual, somatomotor, dorsal attention, ventral attention, limbic, frontoparietal, and default mode networks). In the visual, somatomotor, default mode and ventral attention networks, SC significantly predict FC variance (p < 0.05, 1000 spin test permutations, *Figure 2C*). The visual and somatomotor networks had higher coupling values than the other networks (p < 0.05, Kruskal–Wallis ANOVA, *Figure 2C*). We further investigated the alignment between SC–FC coupling and three fundamental properties of brain organization: evolution expansion (*Hill et al., 2010*), myelin content (*Glasser and Van Essen, 2011*), and functional principal gradient (*Margulies et al., 2016*). Our findings reveal a negative association between regional distribution of SC–FC coupling and evolution expansion (Spearman's $r = -0.52$, p < 0.001, 1000 spin test permutations, *Figure 2D*), as well as with the functional principal gradient (Spearman's $r = -0.46$, p < 0.001, 1000 spin test permutations, *Figure 2F*). Conversely, nodes exhibiting higher SC–FC coupling tended to exhibit higher myelin content (Spearman's $r = 0.49$, p < 0.001, 1000 spin test permutations, *Figure 2E*). In addition, the coupling pattern based on other models (using only MPC or only SCs to predict FC) and the comparison between the models are shown in *Figure 2—figure supplement 1A–C*.

Additionally, we applied Haufe's inversion transform (*Haufe et al., 2014*) to yield predictor weights of various SCs, where higher or lower values indicate stronger positive or negative correlations with FC. Our results demonstrated that different SCs had preferential contributions to FC variance across cortical regions to explain FC variance (p < 0.05, false discovery rate (FDR) corrected, Kruskal–Wallis ANOVA, *Figure 2G*). Specifically, in the MPC, regions with positive correlation were the orbital gyrus, precentral gyrus, right middle temporal gyrus, and temporoparietal junction, while regions with negative correlations were the left superior frontal gyrus, inferior parietal lobule, and bilateral cingulate gyrus. Regarding WMC communication models, the communicability and flow graphs tended to stronger higher positive correlations in the visual, limbic, and default mode networks, whereas the

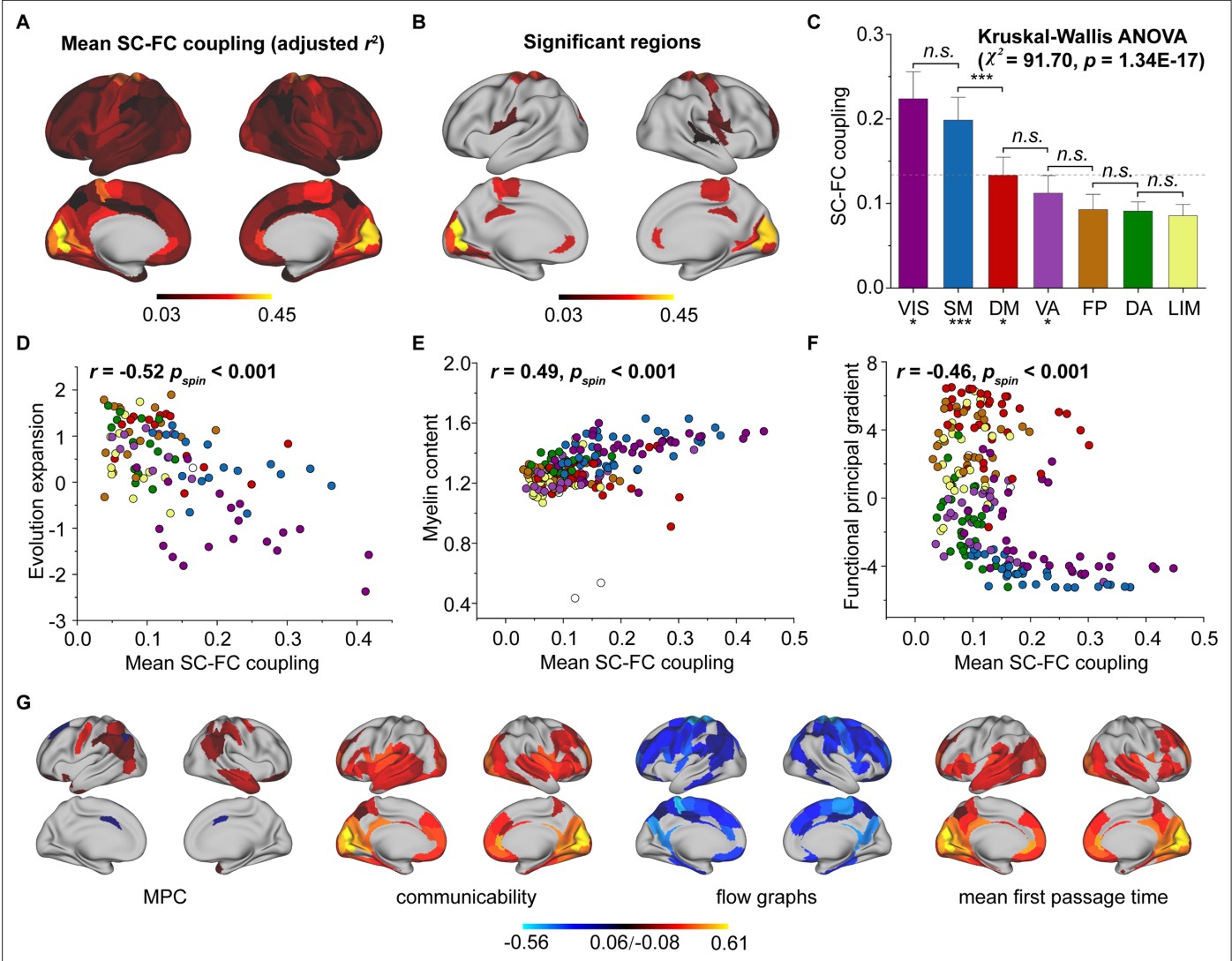

**Figure 2.** Cortical structural connectome–functional connectome (SC–FC) coupling in young individuals. (**A**) Spatial pattern of SC–FC coupling. (**B**) Spatial patterns with significant predictions (p < 0.05, spin test). (**C**) SC–FC coupling comparisons among functional networks. The error bars represent 95% confidence intervals (n = 210). (**D–F**) SC–FC coupling aligns with evolution expansion, myelin content, and functional principal gradient. (**G**) Preferential contributions of cortical regions across different structural connections. Note: ***p < 0.001; *p < 0.05; *n.s.*: p > 0.05. VIS, visual network; SM, somatomotor network; DA, dorsal attention network; VA, ventral attention network; LIM, limbic network; FP, frontoparietal network; DM, default mode network.

The online version of this article includes the following figure supplement(s) for figure 2:

**Figure supplement 1.** Comparison results between different models.

mean first-passage time had stronger negative correlations in the somatomotor, limbic, and fronto-parietal networks.

## Age-related changes in SC–FC coupling with development

To track changes in SC–FC coupling during development, we used a general linear model to assess the effect of age on nodal SC–FC coupling, while controlling for sex, intracranial volume, and in-scanner head motion. Our results revealed that the whole-cortex average coupling increased during development ($\beta_{age}$ = 1.05E−03, F = 3.76, p = 1.93E−04, r = 0.20, p = 3.20E−05, *Figure 3A*). Regionally, the SC–FC coupling of most cortical regions increased with age (p < 0.05, FDR corrected, *Figure 3B*), particularly that in the frontal lobe, middle temporal gyrus, inferior temporal gyrus, parietal lobe, cingulate gyrus, and lateral occipital cortex. Conversely, cortical regions with significantly

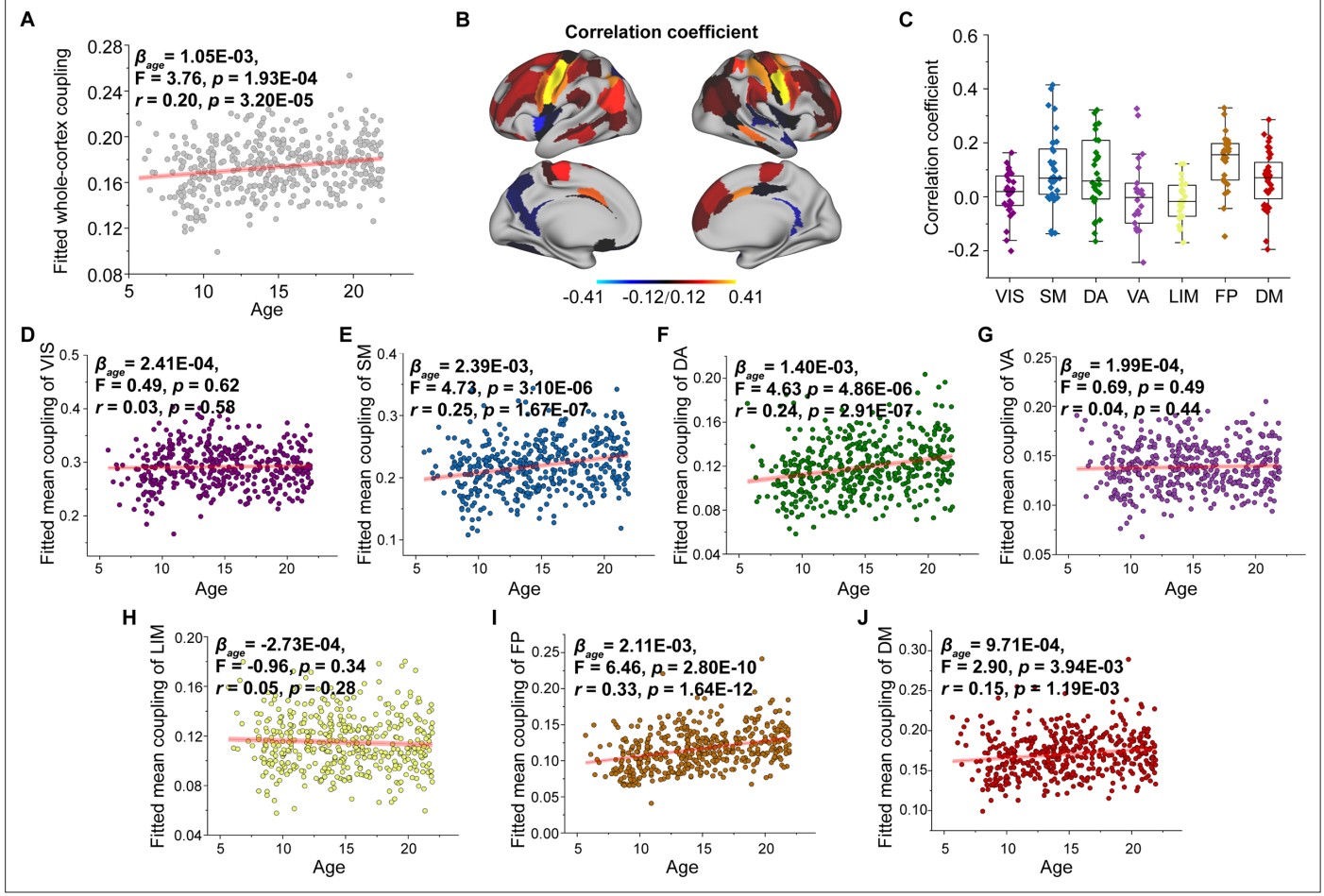

**Figure 3.** Age-related changes in structural connectome–functional connectome (SC–FC) coupling. (**A**) Increases in whole-brain coupling with age. (**B**) Correlation of age with SC–FC coupling across all regions and significant regions (p < 0.05, FDR corrected). (**C**) Comparisons of age-related changes in SC–FC coupling among functional networks. Each point represents a brain region (n = 210). The boxes show the median and interquartile range (IQR; 25–75%), and the whiskers depict 1.5 × IQR from the first or third quartile. (**D–J**) Correlation of age with SC–FC coupling across the VIS, SM, DA, VA, LIM, FP, and DM. VIS, visual network; SM, somatomotor network; DA, dorsal attention network; VA, ventral attention network; LIM, limbic network; FP, frontoparietal network; DM, default mode network.

The online version of this article includes the following figure supplement(s) for figure 3:

**Figure supplement 1.** Age-related changes in microstructure profile covariance (MPC) weight.

**Figure supplement 2.** Age-related changes in communicability weight.

**Figure supplement 3.** Age-related changes in flow graph weight.

**Figure supplement 4.** Age-related changes in the weight of the mean first-passage time.

decreased SC–FC coupling (p < 0.05, FDR corrected, *Figure 3B*) were located in left orbital gyrus, left precentral gyrus, right superior and inferior temporal gyrus, left fusiform gyrus, left superior parietal lobule, left postcentral gyrus, insular gyrus, and cingulate gyrus. Age correlation coefficients distributed within functional subnetworks are shown in *Figure 3C*. Regarding mean SC–FC coupling within functional subnetworks, the somatomotor ($\beta_{age}$ = 2.39E−03, F = 4.73, p = 3.10E−06, r = 0.25, p = 1.67E−07, *Figure 3E*), dorsal attention ($\beta_{age}$ = 1.40E−03, F = 4.63, p = 4.86E−06, r = 0.24, p = 2.91E−07, *Figure 3F*), frontoparietal ($\beta_{age}$ = 2.11E−03, F = 6.46, p = 2.80E−10, r = 0.33, p = 1.64E−12, *Figure 3I*) and default mode ($\beta_{age}$ = 9.71E−04, F = 2.90, p = 3.94E−03, r = 0.15, p = 1.19E−03, *Figure 3J*) networks significantly increased with age and exhibited greater increase. No significant correlations were found between developmental changes in SC–FC coupling and the fundamental properties of cortical organization. Additionally, weights of different SCs varied with age, showing that MPC weight was positively correlated with age and that the weights of WMC

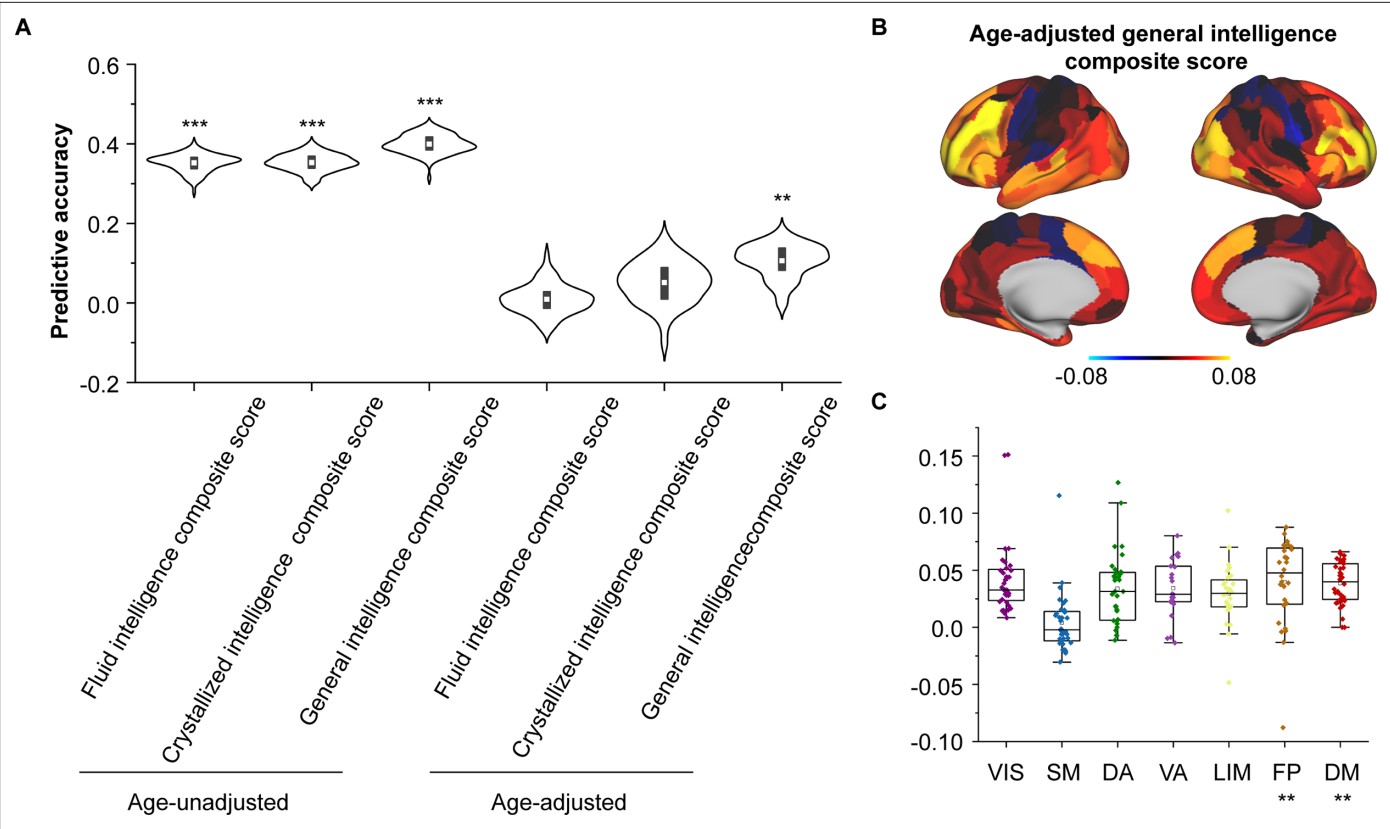

**Figure 4.** Encoding individual differences in intelligence using regional structural connectome–functional connectome (SC–FC) coupling. (**A**) Predictive accuracy of fluid, crystallized, and general intelligence composite scores. (**B**) Regional distribution of predictive weight. (**C**) Predictive contribution of functional networks. Each point represents a brain region (n = 210). The boxes show the median and interquartile range (IQR; 25–75%), and the whiskers depict the 1.5 × IQR from the first or third quartile. Note: ***p < 0.001; **p < 0.01.

The online version of this article includes the following figure supplement(s) for figure 4:

**Figure supplement 1.** Predictive accuracy of regional structural connectome–functional connectome (SC–FC) coupling across cognitive measures.

communication models were stable (*Figure 3—figure supplements 1–4*). The age-related patterns of SC–FC coupling based other coupling models were shown in *Figure 2—figure supplement 1D–F*.

## SC–FC coupling predicts individual differences in cognitive functions

As we found that SC–FC coupling can encode brain maturation, we next evaluated the implications of coupling for individual cognition using Elastic-Net algorithm (*Feng et al., 2022*). After controlling for sex, intracranial volume and in-scanner head motion, we found the SC–FC coupling significantly predicted individual differences in fluid, crystal, and general intelligence (Pearson's *r* = 0.3–0.4, p < 0.001, FDR corrected, *Figure 4A*). Furthermore, even after controlling for age, SC–FC coupling remained a significant predictor of general intelligence better than at chance (Pearson's *r* = 0.11 ± 0.04, p = 0.01, FDR corrected, *Figure 4A*). For fluid and crystal intelligence, the predictive performances of SC–FC coupling were not better than at chance (*Figure 4A*). The predictive performances for other cognitive subscores are shown in *Figure 4—figure supplement 1*. To identify the regions with the greatest contributions to individual differences in age-adjusted general intelligence, we utilized Haufe's inversion transform (*Haufe et al., 2014*) to extract predictor weights across various regions. Our analysis revealed that SC–FC coupling within the prefrontal, temporal, and lateral occipital lobes was the most predictive of individual differences in general intelligence (*Figure 4B*). In addition, we found that the weights of frontoparietal and default mode networks significantly contributed to the prediction of the general intelligence (p < 0.01, 1000 spin test permutations, *Figure 4C*).

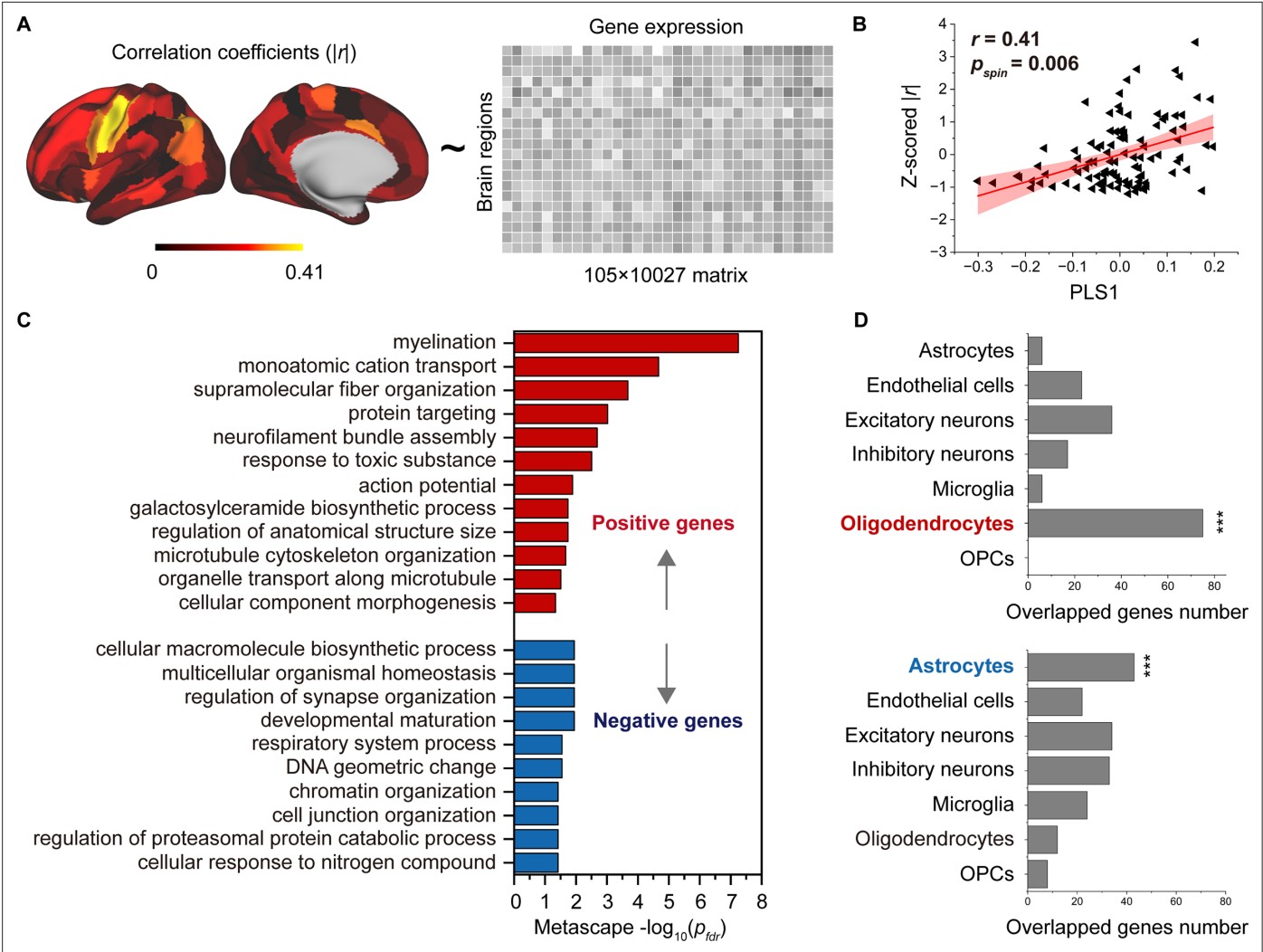

**Figure 5.** Association between developmental changes in structural connectome–functional connectome (SC–FC) coupling and gene transcriptional profiles. (**A**) The map of developmental changes (absolute value of correlation coefficients) in SC–FC coupling across 105 left brain regions (left panel), and the normalized gene transcriptional profiles containing 10,027 genes in 105 left brain regions (right panel). (**B**) The correlation between developmental changes in SC–FC coupling and the first partial least square component (PLS1) from the PLS regression analysis. (**C**) Enriched terms of significant genes. (**D**) Cell type-specific expression of significant genes. Note: $p_{spin}$: spin test; $p_{fdr}$: FDR corrected; ***$p < 0.001$.

The online version of this article includes the following figure supplement(s) for figure 5:

**Figure supplement 1.** Cell-specific expression in each pathway.

## Transcriptomic and cellular architectures of SC–FC coupling development

We employed partial least square (PLS) analysis (**Krishnan et al., 2011**) to establish a link between the spatial pattern of SC–FC coupling development and gene transcriptomic profiles (**Figure 5A**) obtained from the AHBA using a recommended pipeline (**Arnatkeviciute et al., 2019**). The gene expression score of the first PLS component (PLS1) explained the most spatial variance, at 22.26%. After correcting for spatial autocorrelation (**Vos de Wael et al., 2020**), we found a positive correlation (Pearson's $r = 0.41$, $p = 0.006$, 10,000 spin test permutations, **Figure 5B**) between the PLS1 score of genes and the spatial pattern of SC–FC coupling development. In addition, we identified potential transcriptomic architectures using a Gene Ontology (GO) enrichment analysis of biological processes and pathway (**Zhou et al., 2019**), analysing the significant positive and negative genes in PLS1. The positive weight genes (364 genes) were prominently enriched for 'myelination', 'monoatomic cation transport', 'supramolecular fibre organization', etc. ($p < 0.05$, FDR corrected, **Figure 5C**). The negative

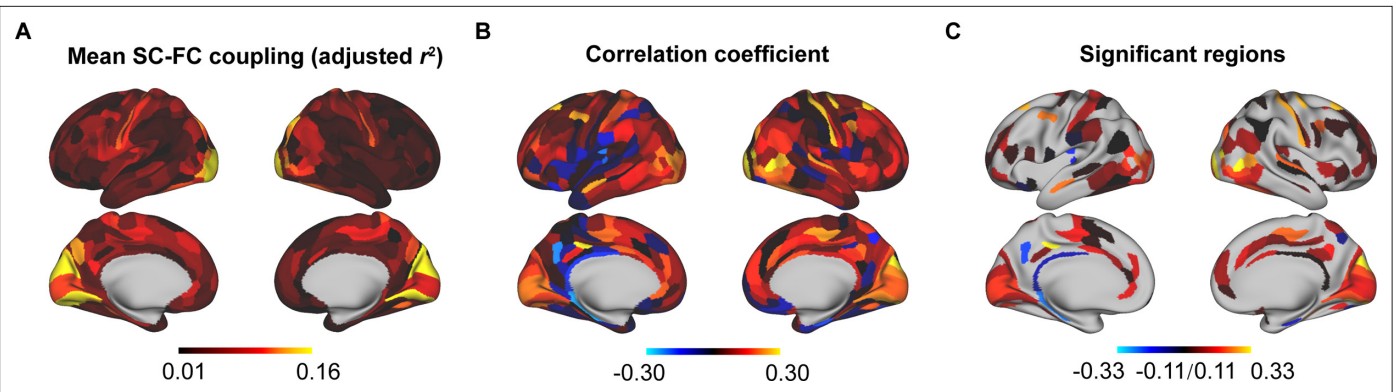

**Figure 6.** Reproducibility analyses with different parcellation templates (HCPMMP). (**A**) Spatial pattern of structural connectome–functional connectome (SC–FC) coupling. (**B**) Correlation of age with SC–FC coupling. (**C**) Correlation of age with SC–FC coupling across significant regions (p < 0.05, FDR corrected).

correlation genes (456 genes) were relatively weakly enriched in 'cellular macromolecule biosynthetic process' and other pathways (p < 0.05, FDR corrected, *Figure 5C*).

To further investigate cell-specific expression patterns associated with SC–FC coupling development, the selected genes in the AHBA were agglomerated into seven canonical cell classes (*Zhang et al., 2016*; *Lake et al., 2018*; *Habib et al., 2017*; *Darmanis et al., 2015*; *Li et al., 2018*; *Seidlitz et al., 2020*): astrocytes, endothelial cells, excitatory neurons, inhibitory neurons, microglia, oligodendrocytes, and oligodendrocyte precursors (OPCs). Our findings showed that the genes with positive weights were significantly expressed in oligodendrocytes (75 genes, p < 0.001, permutation test, *Figure 5D*). The genes with negative weights were expressed in astrocytes (43 genes, p < 0.001, permutation test, *Figure 5D*). Additionally, genes enriched in positive pathways were intensively overexpressed in oligodendrocytes, while genes enriched in three negative pathways were expressed in astrocytes, inhibitory neurons and microglia (p < 0.05, permutation test, *Figure 5—figure supplement 1*).

## Reproducibility analyses different parcellation templates

To evaluate the robustness of our findings to different parcellation templates, using the multimodal parcellation from the Human Connectome Project (HCPMMP) (*Glasser et al., 2016*), we repeated the analyses of the cortical patterns of SC–FC coupling, correlation of age with SC–FC coupling, and gene weights. We observed a similar distribution in SC–FC coupling in which visual and somatomotor networks had higher coupling values than other networks (*Figure 6A*). The SC–FC coupling of most cortical regions increased with age (*Figure 6B*), and the significant regions were similar to those in the main findings (*Figure 6C*, p < 0.05, FDR corrected). The gene weights of HCPMMP was consistent with that of BNA (r = 0.25, p < 0.001).

## Different tractography strategies

To evaluate the sensitivity of our results to tractography strategies, we reconstructed fibres using deterministic tractography with a ball-and-stick model and generated a fibre number-weighted network for each participant. This same pipeline was employed for subsequent SC–FC coupling, prediction, and gene analyses. These two tractography strategies yielded similar findings, as indicated by significant correlations in the mean SC–FC coupling (r = 0.85, p < 0.001, spin test, *Figure 7A*), the correlation of between age and SC–FC coupling (r = 0.79, p < 0.001, spin test, *Figure 7B*), predictive weights on the general intelligence (r = 0.85, p < 0.001, spin test, *Figure 7C*), and gene weights (r = 0.80, p < 0.001, *Figure 7D*).

## Split-half validation

To assess the reproducibility of our findings, we performed a split-half independent validation using the whole dataset (WD). Specifically, we randomly partitioned WD into two independent subsets (S1 and S2), and this process was repeated 1000 times to mitigate any potential bias due to data

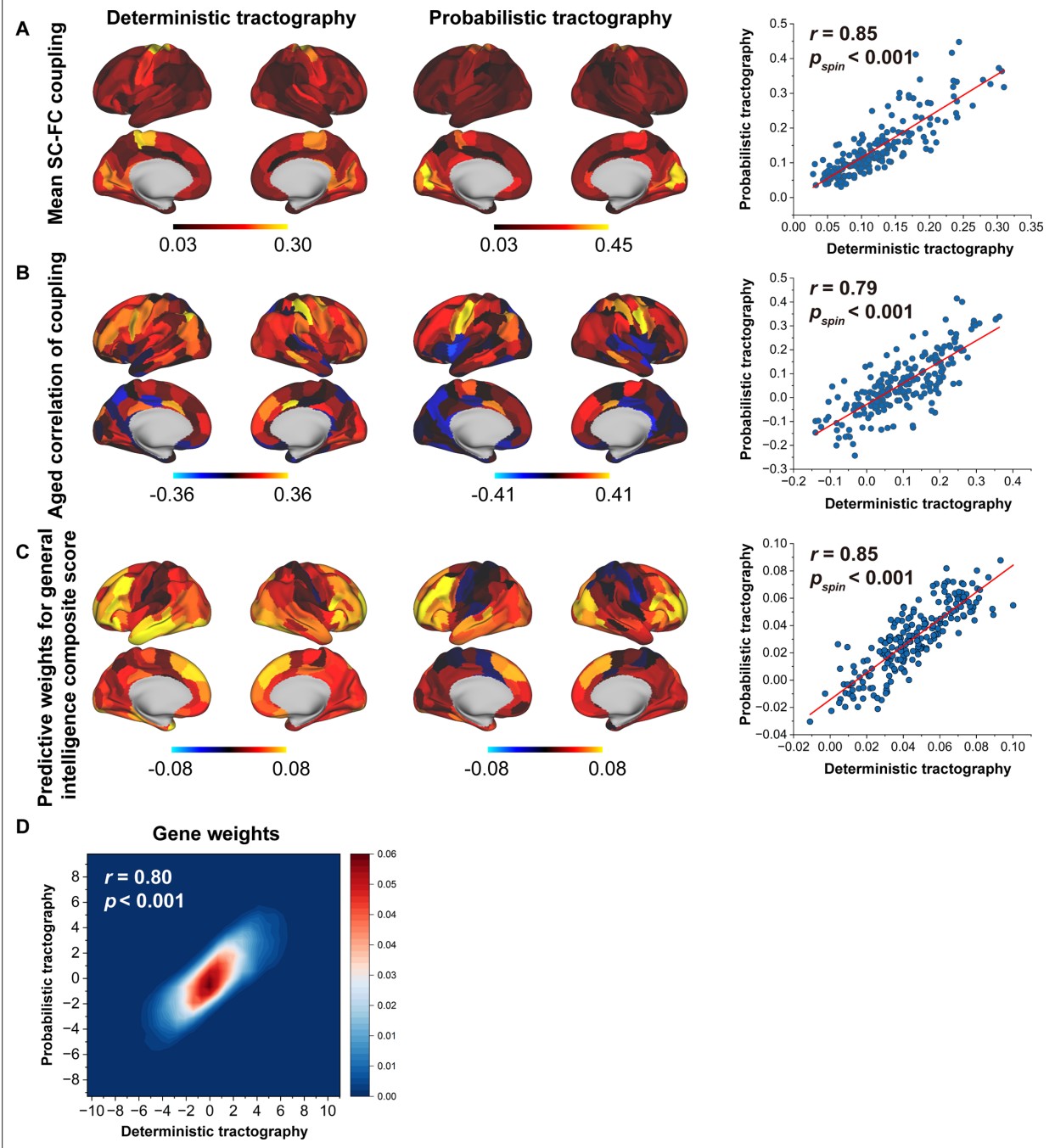

**Figure 7.** Reproducibility analyses with different tractography strategies. (**A**) The consistency of mean structural connectome–functional connectome (SC–FC) coupling between deterministic and probabilistic tractography. (**B**) The consistency of the correlation between age and SC–FC coupling between deterministic and probabilistic tractography. (**C**) The consistent predictive weights for the general intelligence composite score between deterministic and probabilistic tractography. (**D**) The consistency of gene weights between deterministic and probabilistic tractography.

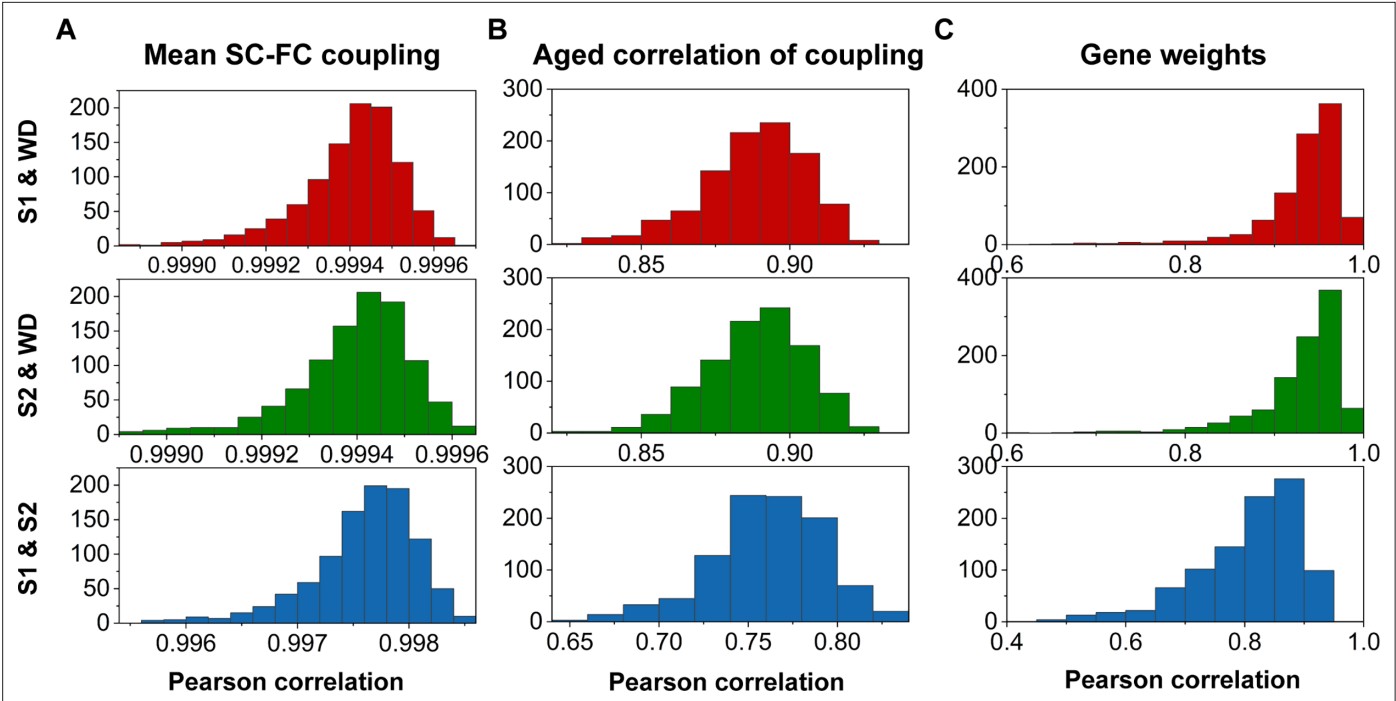

**Figure 8.** Reproducibility analyses with split-half validation. (**A**) The consistency of mean structural connectome–functional connectome (SC–FC) coupling among S1, S2, and the whole dataset (WD). (**B**) The consistency of the correlation between age and SC–FC coupling among S1, S2, and the WD. (**C**) Consistent gene weights among S1, S2, and WD.

partitioning. We then quantified SC–FC coupling, correlation between age and SC–FC coupling, and gene weights in S1 and S2 using the same procedures. Remarkably, we observed high levels of agreement among the datasets (S1, S2, and the WD) as demonstrated in *Figure 8*.

## Discussion

In the present study, we characterized alterations of SC–FC coupling of brain connectome during development by combining intra- and extracortical SC to predict FC based on the HCP-D dataset. We observed that SC–FC coupling was stronger in the visual and somatomotor networks than in other networks, and followed fundamental properties of cortical organization. With development, SC–FC coupling exhibited heterogeneous changes in cortical regions, with significant increases in the somatomotor, frontoparietal, dorsal attention, and default mode networks. Furthermore, we found that SC–FC coupling can predict individual differences in general intelligence, mainly with the frontoparietal and default mode networks contributing higher weights. Finally, we demonstrated that the spatial heterogeneity of changes in SC–FC coupling with age was associated with transcriptomic architectures, with genes with positive weights enriched in oligodendrocyte-related pathways and genes with negative weights expressed in astrocytes. Together, these findings characterized the spatial and temporal pattern of SC–FC coupling of brain connectome during development and the heterogeneity in the development of SC–FC coupling is associated with individual differences in intelligence and transcriptomic architecture.

Intracortical microcircuits are interconnected through extracortical WM connections, which give rise to richly patterned functional networks (*Vázquez-Rodríguez et al., 2019*; *Demirtaş et al., 2019*). Despite extensive research on this topic, the relationship between SC and FC remains unclear. Although many studies have attempted to directly correlate FC with the WMC, this correspondence is far from perfect due to the presence of polysynaptic (indirect) structural connections and circuit-level modulation of neural signals (*Sarwar et al., 2021*; *Baum et al., 2020*; *Honey et al., 2009*; *Damoiseaux and Greicius, 2009*). Biological models can realistically generate these complex structural interconnections, but they have significant temporal and spatial complexity when solving for model parameters (*Wang et al., 2019*; *Woolrich and Stephan, 2013*; *Messé et al., 2015*; *Honey et al., 2007*).

Communication models using the WMC integrate the advantages of different communication strategies and are easy to construct (*Avena-Koenigsberger et al., 2017*). As there are numerous communication models, we identified an optimal combination consisting of three decentralized communication models based on predictive significance: communicability, mean first-passage times of random walkers and flow graphs. We excluded a centralized model (shortest paths), which was not biologically plausible since it requires global knowledge of the shortest path structure (*Zamani Esfahlani et al., 2022*; *Goñi et al., 2014*; *Avena-Koenigsberger et al., 2019*). In our study, we excluded the Euclidean and geodesic distance because spatial autocorrelation is inhibited. This study provides a complementary perspective (in addition to the role of WMC in shaping FC) that emphasizes the importance of intrinsic properties within intracortical circuit in shaping the large-scale functional organization of the human cortex. MPC can link intracortical circuits variance at specific cortical depths from a graph-theoretical perspective, enabling reflection of intracortical microcircuit differentiation at molecular, cellular, and laminar levels (*Valk et al., 2022*; *Paquola et al., 2019b*; *Liu et al., 2022*; *Paquola and Hong, 2023*; *Park et al., 2022*). Coupling models that incorporate these microarchitectural properties yield more accurate predictions of FC from SC (*Demirtaş et al., 2019*; *Deco et al., 2021*).

SC–FC coupling may reflect anatomical and functional hierarchies. SC–FC coupling in association areas, which have lower structural connectivity, was lower than that in sensory areas. This configuration effectively releases the association cortex from strong structural constraints imposed by early activity cascades, promoting higher cognitive functions that transcend simple sensorimotor exchanges (*Buckner and Krienen, 2013*). A macroscale functional principal gradient (*Margulies et al., 2016*; *Huntenburg et al., 2018*) in the human brain has been shown to align with anatomical hierarchies. Our study revealed a similar pattern, where SC–FC coupling was positively associated with evolutionary expansion and myelin content, and negatively associated with functional principal gradient during development. These findings are consistent with previous studies on WMC–FC (*Baum et al., 2020*) and MPC–FC coupling (*Valk et al., 2022*). Notably, we also found that the coupling pattern differed from that in adults, as illustrated by the moderate coupling of the sensorimotor network in the adult population (*Gu et al., 2021*). SC–FC coupling is dynamic and changes throughout the lifespan (*Zamani Esfahlani et al., 2022*), particularly during adolescence (*Valk et al., 2022*; *Baum et al., 2020*), suggesting that perfect SC–FC coupling may require sufficient structural descriptors. Moreover, our results suggested that regional preferential contributions across different SCs lead to variations in the underlying communication process. Interestingly, the two extremes of regions in terms of MPC correlations corresponded to the two anchor points of the gradient (*Paquola et al., 2019a*). The preferential regions in WM communication models were consistent with the adult results (*Zamani Esfahlani et al., 2022*).

In addition, we observed developmental changes in SC–FC coupling dominated by a positive increase in cortical regions (*Baum et al., 2020*), broadly distributed across somatomotor, frontoparietal, dorsal attention, and default mode networks (*Baum et al., 2020*). In a lifespan study, the global SC–FC coupling alterations with age were driven by reduced coupling in the sensorimotor network (*Zamani Esfahlani et al., 2022*). This finding is consistent across age ranges, indicating that sensorimotor coupling changes appear throughout development and ageing. Furthermore, we investigated the relationships of coupling alterations with evolutionary expansion and functional principal gradient but found no significant correlations, in contrast to a previous study (*Baum et al., 2020*). These discrepancies likely arise from differences in coupling methods. We also found the SC–FC coupling with age across regions within subnetworks has more variability than the differences between networks, suggesting that the coupling with age is more likely region-dependent than network-dependent.

The neural circuits in the human brain support a wide repertoire of human behaviour (*Chen et al., 2022*). Our study demonstrates that the degree of SC–FC coupling in cortical regions can significantly predict cognitive scores across various domains, suggesting that it serves as a sensitive indicator of brain maturity. Moreover, even after controlling for age effects, SC–FC coupling significantly predicted general intelligence, suggesting that it can partly explain individual differences in intelligence, as shown in previous studies (*Gu et al., 2021*). In another study (*Baum et al., 2020*), positive correlations between executive function and SC–FC coupling were mainly observed in the rostro-lateral frontal and medial occipital regions, whereas negative associations were found in only the right primary motor cortex. While SC–FC coupling was not found to predict age-adjusted executive function in our study, we observed that the frontoparietal network and the default mode network specifically

contributed higher positive prediction weights for general intelligence, whereas the somatomotor network had negative prediction weights (*Gu et al., 2021*). The maturation of the frontoparietal network and default mode network continues into early adulthood, providing an extended window for the activity-dependent reconstruction of distributed neural circuits in the cross-modal association cortex (*Buckner and Krienen, 2013*). As we observed increasing coupling in these networks, this may have contributed to the improvements in general intelligence, highlighting the flexible and integrated role of these networks.

Classic twin studies have reported that the heritability of coupling differs among cortical regions, with higher heritability in the visual network than in other cortical networks (*Gu et al., 2021*). An inverse correlation between the pattern of SC–FC coupling and heritable connectivity profiles has been reported (*Valk et al., 2022*). This led us to hypothesize that the development of SC–FC coupling may be influenced by the expression patterns of the genetic transcriptome across various cell types with different spatial distributions. Our findings suggest that the spatial development of SC–FC coupling is associated with underlying transcriptome structure. Specifically, genes positively associated with the development of SC–FC coupling were enriched in oligodendrocyte-related pathways. Oligodendrocytes, specialized glial cells in the central nervous system, play a crucial role in myelination by producing myelin sheaths that enable saltatory conduction and provide metabolic support to axons (*Simons and Nave, 2015*). Defects in myelination have been linked to developmental disorders (*Berry et al., 2020*). This seems to indicate that significant alterations in SC–FC coupling during development may reflect neural plasticity, such as activity-dependent myelination of axons connecting functionally coupled regions (*Gibson et al., 2014*; *Mount and Monje, 2017*). Conversely, we found that genes negatively correlated with SC–FC coupling were enriched in two specific gene pathways within astrocytes, inhibitory neurons and microglia. Both astrocytes and microglia have been implicated in synaptic pruning, a critical developmental process for the formation of fully functional neuronal circuits that eliminates weak and inappropriate synapses (*Kurshan and Shen, 2019*; *Van Horn and Ruthazer, 2019*; *Faust et al., 2021*). Importantly, the precise establishment of synapses is crucial for establishing the intercellular connectivity patterns of GABAergic neurons (*Favuzzi et al., 2019*). These findings suggest that the subtle alterations observed in SC–FC coupling are closely associated with the refinement of mature neural circuits.

Several methodological issues must be addressed. First, we implemented a conservative quality control procedure to address head motion, which unavoidably resulted in the loss of some valuable data. Given the confounding influence of head motion in fMRI studies, especially those involving developing populations, we applied censoring of high-motion frames and included motion as a covariate in the generalized linear model (GLM) analysis and cognitive prediction to minimize its effects (*Zamani Esfahlani et al., 2022*; *Chen et al., 2022*; *Ciric et al., 2017*; *Li et al., 2022*). Second, although we observed SC–FC coupling across development by integrating intra- and extracortical SC to predict FC, it is worth noting that combining deep learning models (*Sarwar et al., 2021*), biophysical models (*Breakspear, 2017*; *Sanz-Leon et al., 2015*), or dynamic coupling (*Demirtaş et al., 2019*; *Liu et al., 2022*) perspectives may provide complementary insights. Third, the appropriateness of structurally defined regions for the functional analysis is also a topic of important debate. Fourth, we focused solely on cortico-cortical pathways, excluding subcortical nuclei from analysis. This decision stemmed from the difficulty of reconstructing the surface of subcortical regions (*Glasser et al., 2013*) and characterizing their connections using MPC technique, as well as the challenge of accurately resolving the connections of small structures within subcortical regions using whole-brain diffusion imaging and tractography techniques (*Thomas et al., 2014*; *Reveley et al., 2015*). In addition, the reconstruction of short connections between hemispheres is a notable challenge. Fifth, it is important to acknowledge that changes in gene expression levels during development may introduce bias in the results. Finally, validation of sensitivity across independent datasets is a crucial step in ensuring the reliability of our results. To address this, we employed an alternative split-half validation strategy and the results supported the reliability of the current findings. However, future verification of current findings on independent datasets are still needed.

## Conclusions

Overall, this study sheds light on the development of SC–FC coupling in the brain and its relationship to cognitive function and gene expression patterns. The results improve our understanding of

the fundamental principles of brain development and provide a basis for future research in this area. Further investigations are needed to fully explore the clinical implications of SC–FC coupling for a range of developmental disorders.

## Materials and methods

### Participants

We selected 439 participants (207 males, mean age = 14.8 ± 4.2 years, age range = [5.7, 21.9]) from the HCP-D Release 2.0 data (https://www.humanconnectome.org/study/hcp-lifespan-development) after conducting rigorous checks for data completeness and quality control. The HCP-D dataset comprised 652 healthy participants who underwent multimodal MRI scans and cognitive assessments, and the detailed inclusion and exclusion criteria for this cohort have been described in *Somerville et al., 2018*. All participants or their parents (for participants under the age of 18 years) provided written informed consent and assent. The study was approved by the Institutional Review Board of Washington University in St. Louis.

### Imaging acquisition

The MRI data were obtained with a Siemens 3T Prisma with a 32-channel phased array head coil, and detailed imaging parameters are available in *Harms et al., 2018*. High-resolution T1w images were acquired using a 3D multiecho MPRAGE sequence (0.8 mm isotropic voxels, repetition time (TR)/ inversion time (TI) = 2500/1000 ms, echo time (TE) = 1.8/3.6/5.4/7.2 ms, flip angle = 8°, up to 30 reacquired TRs). The structural T2w images were collected with a variable-flip-angle turbo-spin-echo 3D SPACE sequence (0.8 mm isotropic voxels, TR/TE = 3200/564 ms, up to 25 reacquired TRs). The dMRI scans included four consecutive runs with a 2D 4×multiband spin−echo echo-planar imaging (EPI) sequence (1.5 mm isotropic voxels, 185 diffusion directions with $b$ = 1500/3000 s/mm$^2$ and 28 $b$ = 0 s/mm$^2$ volumes, TR = 3.23 s, flip angle = 78°). The rs-fMR images were acquired using a 2D 8×multiband gradient-recalled echo EPI sequence (2.0 mm isotropic voxels, TR/TE = 800/37 ms, flip angle = 52°). Each rs-fMRI scan duration was 26 min (four runs of 6.5 min) for participants over 8 years old and 21 min (six runs of 3.5 min) for participants who were 5–7 years old.

### Imaging preprocessing

All structural, diffusion, and functional images underwent minimal preprocessing (*Glasser et al., 2013*). We specifically processed dMRI data referring to the publicly available code from https://github.com/Washington-University/HCPpipelines, *Brown et al., 2024* since the HCP-D has not released preprocessed dMRI results. Briefly, structural T1w and T2w images went through gradient distortion correction, alignment, bias field correction, registration to Montreal Neurological Institute (MNI) space, WM and pial surface reconstruction, segment structures, and surface registration and downsampling to 32 k_fs_LR mesh. A T1w/T2w ratio image, which indicates intracortical myelin, was produced for each participant (*Glasser and Van Essen, 2011*). The BNA (*Fan et al., 2016*) was projected on native space according to the official scripts (http://www.brainnetome.org/resource/) and the native BNA was checked by visual inspection. Regarding fMRI data, the preprocessing pipeline included spatial distortion correction, motion correction, EPI distortion correction, registration to MNI space, intensity normalization, mapping volume time series to 32 k_fs_LR mesh, and smoothing using a 2-mm average surface vertex. Following our previous methodological evaluation study (*Feng et al., 2022*), the dMRI procedures consisted of intensity normalization of the mean $b_0$ image, correction of EPI distortion and eddy current, motion correction, gradient nonlinearity correction, and linear registration to T1w space.

### Network computation MPC

The MPC can capture cytoarchitectural similarity between cortical areas (*Paquola et al., 2019b*). We first reconstructed 14 cortical surfaces from the WM to the pial surface using a robust equivolumetric model (*Paquola et al., 2019b*; *Waehnert et al., 2014*). Then, the T1w/T2w ratio image was used to sample intracortical myelin intensities at these surfaces. We averaged the intensity profiles of vertices over 210 cortical regions according to the BNA (*Fan et al., 2016*). Finally, we computed pairwise partial correlations between regional intensity profiles, while controlling for the average intensity

profile. After removing negative correlations, we used Fisher's *r*-to-*z*-transformation to generate an individual MPC.

## White matter connectome

Following our previous methodological evaluation study (***Feng et al., 2022***), the ball-and-stick model estimated from the bedpostx command-line in the FDT toolbox of FSL (https://fsl.fmrib.ox.ac.uk/fsl/fslwiki/FDT) was used to estimate fibre orientations (three fibres modelled per voxel) (***Behrens et al., 2003***; ***Jbabdi et al., 2012***; ***Behrens et al., 2007***; ***Hernández et al., 2013***). The BNA atlas was applied to individual volume space by inverse transformation derived from preprocessed steps. Next, probabilistic tractography (probtrackx) (***Behrens et al., 2007***; ***Hernandez-Fernandez et al., 2019***) was implemented in the FDT toolbox to estimate the probability of connectivity between two regions by sampling 5000 fibres for each voxel within each region, correcting for distance, dividing by the total fibres number in source region, and calculating the average bidirectional probability (***Feng et al., 2022***). Notably, the connections in subcortical areas were removed. A consistency-based thresholding approach (weight of the coefficient of variation at the 75th percentile) was used to remove spurious connections, and retain consistently reconstructed connections across subjects (***Baum et al., 2020***; ***Roberts et al., 2017***).

## Functional network

To further clean the functional signal, we performed frame censoring, regressed out nuisance variables (including WM, cerebrospinal fluid, global signal, and 12 motion parameters), and executed temporal bandpass filtering (0.01–0.1 Hz). Specifically, we identified censored frames with motion greater than 0.15 mm (***Zamani Esfahlani et al., 2022***) based on the Movement_RelativeRMS.txt file. We flagged one frame before and two frames after each censored frame, along with any uncensored segments of fewer than five contiguous frames, as censored frames as well (***Li et al., 2022***). We discarded fMRI runs with more than half of the frames flagged as censored frames, and excluded participants with fewer than 300 frames (less than 4 min). The nuisance variables were removed from time series based on general linear model. We averaged the time series of vertices into 210 cortical regions according to the BNA (***Fan et al., 2016***). We then computed pairwise Pearson's correlations between regional time series, and applied Fisher's *r*-to-*z*-transformation to the resulting correlations to generate individual FC.

## Communication model

Twenty-seven communication models (***Zamani Esfahlani et al., 2022***) were subsequently derived from the WMC, defined as follows:

### Shortest path length

The connectivity of network can be associated with cost, in which higher connectivity strength has lower cost. Let there be a source node $s$, and a target node $t$, $p_{s \to t} = \{p_{si}, p_{ij}, \ldots, p_{kt}\}$ is the sequence of paths between $s$ and $t$. Here, a transformation strategy $tp_{si} = p_{ij}^{-\gamma}$ is used to obtain the $tp_{s \to t} = \{tp_{si}, tp_{ij}, \ldots, tp_{kt}\}$. The shortest path length $sp_{s \to t}$ is calculated as the minimized sum of $tp_{s \to t}$. We set $\gamma = 0.12, 0.25, 0.5, 1, 2$, and 4.

### Communicability

Communicability (***Crofts and Higham, 2009***) is a weighted sum of walks along all connections. The weighted connectivity matrix $\mathbf{A}$ is normalized as $\mathbf{A}' = \mathbf{D}^{-1/2}\mathbf{A}\mathbf{D}^{-1/2}$, where $\mathbf{D}$ is the degree diagonal matrix. The communicability is exponentiated as $\mathbf{G} = e^{\mathbf{A}'}$.

## Cosine similarity

Cosine similarity $c_{st} = \frac{\boldsymbol{n}_s \cdot \boldsymbol{n}_t}{\|\boldsymbol{n}_s\| \cdot \|\boldsymbol{n}_t\|}$ measures the angle between connection patterns of two nodes, $\boldsymbol{n}_s = [n_{s1}, n_{s2}, \ldots, n_{sm}]$ and $\boldsymbol{n}_t = [n_{t1}, n_{t2}, \ldots, n_{tm}]$, where $\|\cdot\|$ is the norm of the vector, and $m$ is the number of brain regions.

## Search information

Search information (**Rosvall et al., 2005**) quantifies the amount of information (in bits) required to traverse shortest paths in a network. If the node sequence of shortest path between $s$ and $t$ is given by $|sp_{s \to t}| = \{s, i, j, \ldots, k, l, t\}$, then the probability of taking that path is given by $B\left(sp_{s \to t}\right) = B_{si} \times B_{ij} \times \ldots \times B_{kl} \times B_{lt}$, where $B_{ij} = \frac{p_{ij}}{\sum_j p_{ij}}$. The information transmitted along this path, is then $si\left(sp_{s \to t}\right) = \log_2\left[B\left(sp_{s \to t}\right)\right]$.

## Matching index

Matching index (**Hilgetag et al., 2000**) is a measure of overlap between pairs of nodes based on their connectivity profiles excluding their mutual connections, here defined as $mi_{ij} = \frac{\sum_{i \neq s,t}\left(p_{si} + p_{it}\right)\theta\left(p_{si}\right)\theta\left(p_{it}\right)}{\sum_{i \neq t} p_{si} + \sum_{i \neq s} p_{it}}$, where $\theta\left(p_{si}\right) = 1$ if $p_{si} > 0$ and 0 otherwise.

## Path transitivity

Path transitivity (**Goñi et al., 2014**) captures the transitivity of the path linking source nodes to a target node or, put differently, the density of local detours that are available along the path. This leads to the definition of 'path transitivity' as $pt_{st} = \frac{2\sum_{i \in sp_{s \to t}} \sum_{j \in sp_{s \to t}} mi_{ij}}{|sp_{s \to t}|\left(|sp_{s \to t}| - 1\right)}$.

## Greedy navigation

Greedy navigation (**Seguin et al., 2018**) is defined as the number of hops in the complete paths revealed by the navigation process. Note that for some node pairs, the navigation procedure leads to a dead end or a cycle—in which case the number of hops is listed as $\infty$.

## Mean first-passage times of random walkers

Mean first-passage times of random walkers (**Noh and Rieger, 2004**) refers to the expected number of steps in a random walk starting at node $s$ to ending at node $t$.

## Flow graphs

Flow graphs (**Lambiotte et al., 2011**) are a transformation of a network's (possibly sparse) connectivity matrix $\mathbf{A}$ into a fully weighted matrix in which the dynamics of a Markov process are embedded into edge weights. For a continuous random walk with dynamics $r_i = -\sum_j L_{ij} r_j$ on node $i$, the corresponding flow graph is given by $g\left(t\right)_{ij} = \left(e^{-t\mathbf{L}}\right)_{ij} s_j$. In these expressions, the matrix $\mathbf{L} = \mathbf{D} - \mathbf{A}/\mathbf{s}$ is the normalized Laplacian, where $s_i = \sum_j A_{ij}$ is a node's degree or weighted degree and $\mathbf{D}$ is the degree diagonal matrix (a square matrix the elements of $s$ along its diagonal), and $g\left(t\right)_{ij}$ represents the probabilistic flow of random walkers between nodes $i$ and $j$ at time $t$. Here, we generated flow graphs using both binary and weighted structural connectivity matrices and evaluated them at different Markov times, $t$. Specifically, we focused on $t = 1, 2.5, 5$, and $10$.

## Quality control

The exclusion of participants in the whole multimodal data processing pipeline is depicted in **Figure 9**. In the context of fMRI data, we computed Pearson's correlation between motion and age, as well as between the number of remaining frames and age, for the included participants aged 5–22 and 8–22 years, respectively. These correlations are presented in **Figure 9—figure supplement 1**.

## Cognitive scores

We included 11 cognitive scores which were assessed with the National Institutes of Health (NIH) Toolbox Cognition Battery (https://www.healthmeasures.net/exploremeasurement-systems/nih-toolbox), including episodic memory, executive function/cognitive flexibility, executive function/

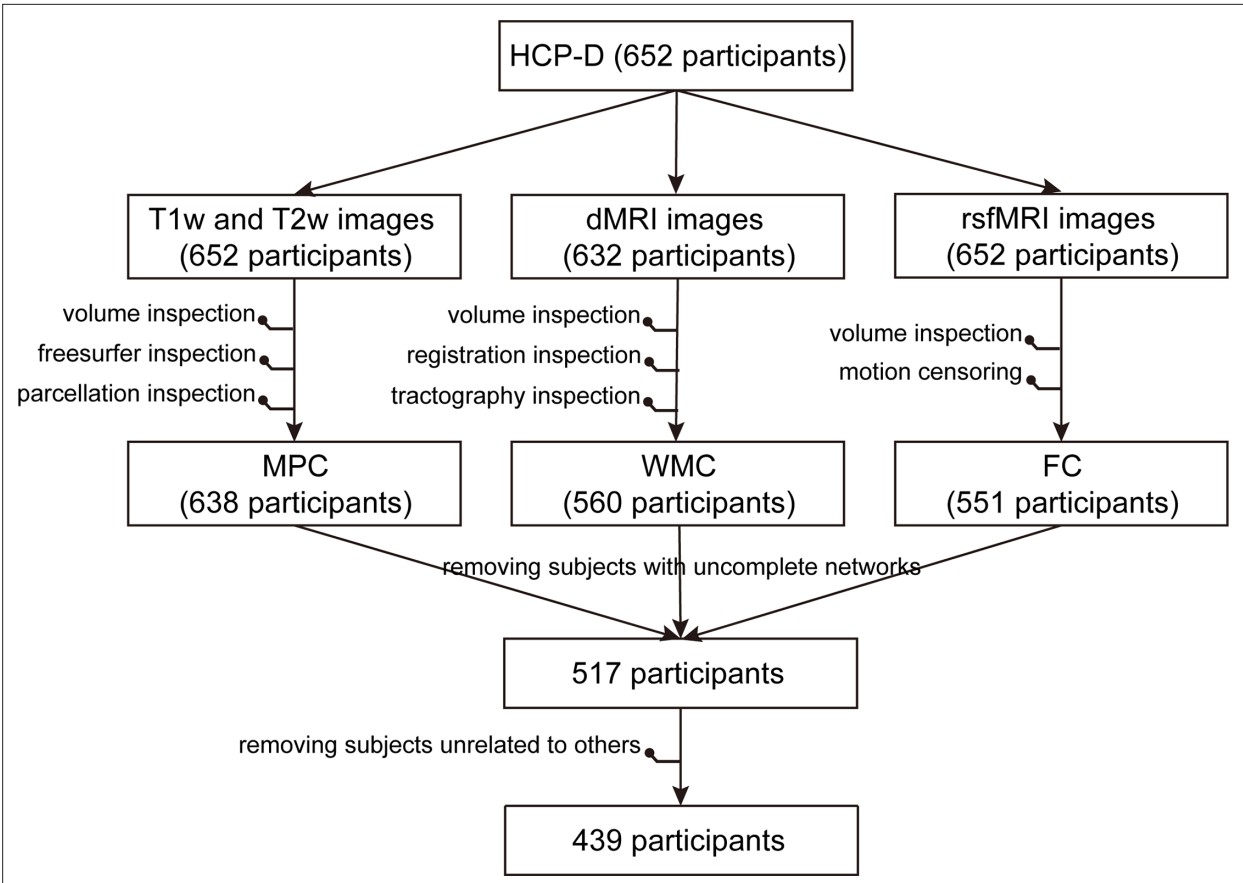

**Figure 9.** Exclusion of participants in the whole multimodal data processing pipeline.

The online version of this article includes the following figure supplement(s) for figure 9:

**Figure supplement 1.** Correlations between motion and age and number of remaining frames and age.

**Figure supplement 2.** Cognitive scores and age distributions of scans.

inhibition, language/reading decoding, processing speed, language/vocabulary comprehension, working memory, fluid intelligence composite score, crystal intelligence composite score, early child intelligence composite score, and total intelligence composite score. Distributions of these cognitive scores and their relationship with age are illustrated in *Figure 9—figure supplement 2*.

## SC–FC coupling

A multilinear model (*Vázquez-Rodríguez et al., 2019*) was constructed to examine the relationship of individual nodewise SC profiles and FC profiles. For a given node, the predictive variable was nodal SC $S = \{s_1, s_2, \cdots, s_i, \cdots, s_n\}$, $s_i \in R^m$ where $s_i$ is the $i$th SC profiles, $n$ is the number of SC profiles, and $m$ is the node number. The nodal functional profile $f$ is the dependent variable.

$$f = b_0 + b_1 s_1 + b_2 s_2 + \cdots + b_i s_i + \cdots + b_n s_n \tag{1}$$

where the intercept $b_0$ and regression coefficients $b_i$ are estimated model parameters. For each participant, goodness of fit per node represents the nodal coupling between SC and FC, quantified as the adjusted coefficient of determination (*Zamani Esfahlani et al., 2022*)

$$R^2_{adjusted} = 1 - \frac{\left(1 - R^2\right)\left(N_c - 1\right)}{N_c - N_p - 1} \tag{2}$$

where $R^2$ is the unadjusted coefficient of determination, $N_c$ is the number of connection ($N_c = 245$ for BNA), and $N_p$ is the number of predictors.

In the present study, WMC communication models that represented diverse geometric, topological, or dynamic factors, were used to explain nodal FC variation. Notably, too many predictors will result in overfitting and blindly increase the explained variance. And covariance structure among the predictors may lead to unreliable predictor weights. Thus, we applied Haufe's inversion transform (*Haufe et al., 2014*) to address these issues and identified reliable communication mechanisms. Specifically, we used all 27 communication models to predict FC at the node level for each participant. We applied Haufe's inversion transform (*Haufe et al., 2014*) to obtain predictor weights for each model, with higher or lower values indicating stronger positive or negative correlations with FC. Next, we generated 1000 FC permutations through a spin test (*Alexander-Bloch et al., 2018*) for each nodal prediction in each subject and obtained random distributions of model weights. These weights were averaged over the group and were investigated the enrichment of the highest weights per region to assess whether the number of highest weights across communication models was significantly larger than that in a random discovery.

The significant communication models were used to represent WMC communication properties and to predict functional profiles in conjunction with MPC as structural profiles (predictors). To test the significance of the resulting adjusted $R^2$ values and system specific of coupling, we generated a null predictive model using a spin test (*Alexander-Bloch et al., 2018*) with 1000 spatially constrained repetitions. We also used Kruskal–Wallis nonparametric one-way analysis of variance (Kruskal–Wallis ANOVA) to compare coupling differences between systems. To investigate the contributions of various structural predictors, we applied Kruskal–Wallis ANOVA to test the predictive weights derived by Haufe's inversion transform, identifying optimal predictors across regions. We corrected for multiple comparisons using FDR correction. Additionally, we used a general linear model to explore age-related developmental patterns of SC–FC coupling, while controlling for sex, intracranial volume, and in-scanner head motion. Similarly, the system-specific significance of coupling alteration was calculated based on the 1000 repetitions of the spin test. In addition, we have constructed the models using only MPC or SCs to predict FC, respectively. Spearman's correlation was used to assess the consistency between spatial patterns based on different models.

We examined the associations of SC–FC coupling and its developmental pattern with evolution expansion (*Hill et al., 2010*), myelin content (*Glasser and Van Essen, 2011*), and functional principal gradient (*Margulies et al., 2016*). Spearman's correlation analyses were used to quantify the strength of correlations, with significance corrected for spatial autocorrelation with 1000 repetitions of the spin test.

## Prediction of cognitive function

Based on our predictive evaluation work (*Feng et al., 2022*), the Elastic-Net algorithm was applied to predict cognitive performance using nodal SC–FC coupling, which tends to yield robust prediction performance across various dimensions of cognitive tasks. The objective function is as follows:

$$L\left(\mathbf{Y}, f\left(\mathbf{X}, \mathbf{w}\right)\right) = \sum_{i=0}^{n} \left(y_i - f\left(x_i\right)\right)^2 + \alpha \sum_{j=1}^{m} \left(\beta \left|w_j\right| + \frac{1}{2}\left(1 - \beta\right) \left|\left|w_j\right|\right|^2\right) \tag{3}$$

where $\mathbf{x} = \left\{x_1, x_2, \ldots, x_n\right\}$ represents an observation set (e.g., SC–FC coupling) with a sample size of $n$, and $\mathbf{y} = \left\{y_1, y_2, \ldots, y_n\right\}$ is a label set (e.g., cognitive measure). The model solves the fitting coefficient $\mathbf{w} = \left(w_1, w_2, \ldots, w_m\right)$ under the minimization objective function $L\left(\mathbf{Y}, f\left(\mathbf{X}, \mathbf{w}\right)\right)$. The $L1$ regularization term $|\cdot|$ and $L2$ regularization term $||\cdot||^2$ constraint the fitting coefficient to ensure model generalization ability. $\alpha$ represents regularization strength, controlling the compression loss scale, and $\beta$ denotes a trade-off parameter between the $L1$ and $L2$ terms.

We employed a nested fivefold cross-validation (CV) framework comprising an external CV and an internal CV (*Feng et al., 2022*). In the external CV, observations were randomly partitioned into fivefolds, with four of them included in the training set used to develop the model and the remaining fold used as a testing set to assess the predictive accuracy of the model. This process was repeated 100 times, and the final model performance was evaluated by averaging the predictive accuracy across the 100 models. In the internal CV, the hyperparameter spaces were first defined as $\alpha \in \left\{x | x = 2^n, n \in \mathbf{Z}, n \in \left[-10, 5\right]\right\}$ and $\beta \in \left\{x | x = 0.1n, n \in \mathbf{Z}, n \in \left[0, 10\right]\right\}$. Then, the training set was further divided into fivefolds. Fourfolds composed the internal training set, which was used to generate models by successively applying 16 × 11 hyperparametric combinations, and the remaining

fold was defined as the validation set and used to find the optimal combination. Subsequently, we retrained the model on the training set using the optimal hyperparametric combination and assessed its predictive performance on the testing set by performing Pearson's correlation analyses of the relationship between the predicted and labelled values.

Prior to applying the nested fivefold CV framework to each behaviour measure, we regressed out covariates including sex, intracranial volume, and in-scanner head motion from the behaviour measure (*Chen et al., 2022*; *Li et al., 2022*). Specifically, we estimated the regression coefficients of the covariates using the training set and applied them to the testing set. This regression procedure was repeated for each fold. Additionally, we conducted control analyses using age-adjusted behavioural measures to investigate the effect of age on the predictive performance of SC–FC coupling.

To evaluate whether our model performed better than at chance on each behaviour measure, we performed 1000 permutation tests by randomly shuffling the behaviour measure across participants, generating a null model of predicted performance using the same procedures. We then used the corrected resampled *t* test to determine statistical significance (*Bouckaert and Frank, 2004*; *Nadeau and Bengio, 2003*). We corrected for multiple comparisons using FDR correction. For model interpretability, we applied Haufe's inversion transform (*Haufe et al., 2014*) to obtain predicted weights for various brain regions. The significance of the weights for each system was assessed by comparing them to those generated by a spin test (*Alexander-Bloch et al., 2018*) with 1000 repetitions.

## Association between alterations of SC–FC coupling and gene expression

We preprocessed the anatomic and genomic information of the AHBA dataset following a recommended pipeline (*Arnatkeviciute et al., 2019*). Specifically, we used FreeSurfer (https://surfer.nmr.mgh.harvard.edu/fswiki/) to generate preprocessed structural data for each donor and projected the BNA template onto native fsaverage space using official scripts (http://www.brainnetome.org/resource/). Finally, we produced an averaged gene expression profile for 10,027 genes covering 105 left cortical regions. Restricting analyses to the left hemisphere will minimize variability across regions (and hemispheres) in terms of the number of samples available (*Arnatkeviciute et al., 2019*).

PLS analysis (*Krishnan et al., 2011*) was performed to mine the linear association between the spatial development pattern of SC–FC coupling and gene expression profiles. We used absolute values of the correlation between age and SC–FC coupling in 105 left cortical regions as predicted variables and the gene expression profiles of the corresponding regions as predictor variables. Pearson's correlation coefficient was calculated to determine the association between the PLS score and the absolute correlation value between age and SC–FC coupling. To correct for spatial autocorrelation, we compared the empirically observed value to spatially constrained null models generated by 10,000 spin permutations (*Alexander-Bloch et al., 2018*). We then transformed the gene weight on PLS1 into a *z* score by dividing the standard deviation of the corresponding weights estimated from bootstrapping, and ranked all genes accordingly. We identified significant genes at a threshold of p < 0.05 and classified them as having positive or negative gene weights. To understand the functional significance of these genes, we performed gene functional enrichment analysis (GO analysis of biological processes and pathways) using Metascape (*Zhou et al., 2019*). We focused on the selected genes with positive or negative weights and retained enrichment pathways with an FDR corrected <0.05.

To investigate the cell type-specific expression of the selected genes, we assigned them to 58 cell types derived from five studies (*Zhang et al., 2016*; *Lake et al., 2018*; *Habib et al., 2017*; *Darmanis et al., 2015*; *Li et al., 2018*) focusing on single-cell research using the human postnatal cortex. To avoid potential bias in cell-type assignment, we grouped these cell types into seven canonical classes: astrocytes, endothelial cells, excitatory neurons, inhibitory neurons, microglia, oligodendrocytes, and OPCs (*Seidlitz et al., 2020*; *Li et al., 2021*). We generated a null model by performing 10,000 random resamplings of genes within each cell type. We then tested the significance of our results against this null model. Additionally, we subjected the genes associated with each enriched term to the same analysis to explore the specificity of the cell type.

## Reproducibility analyses

To evaluate the robustness of our findings under different parcellation templates, we computed MPC, SCs (WMC, communicability *Crofts and Higham, 2009*), mean first-passage times of random walkers

(*Noh and Rieger, 2004*), and flow graphs (timescales = 1), and FC using the multimodal parcellation from the Human Connectome Project (HCPMMP) (*Glasser et al., 2016*). We used the multilinear model to examine the association of individual nodewise SC and FC profiles. Then, a general linear model was used to explore age-related developmental patterns of SC–FC coupling, while controlling for sex, intracranial volume, and in-scanner head motion. We corrected for multiple comparisons using FDR correlation. Finally, we produced an averaged gene expression profile for 10,027 genes covering 176 left cortical regions based on HCPMMP and obtained the gene weights by PLS analysis. We performed Pearson's correlation analyses to assess the consistency of gene weights between HCPMMP and BNA.

To evaluate the sensitivity of our results to deterministic tractography, we used the Camino toolbox (http://camino.cs.ucl.ac.uk/) to reconstruct fibres with a ball-and-stick model estimated from bedpostx results (*Hernández et al., 2013*) and to generate a fibre number-weighted network using the BNA atlas. We then calculated the communication properties of the WMC including communicability, mean first-passage times of random walkers, and flow graphs (timescales = 1). The same pipeline was used for subsequent SC–FC coupling, prediction, and gene analysis. To assess the consistency of our results between deterministic and probabilistic tractography, we performed Pearson's correlation analyses with significance corrected for spatial autocorrelation through 1000 repetitions of the spin test.

To evaluate the generalizability of our findings, we adopted a split-half CV strategy by randomly partitioning the WD into two independent subsets (S1 and S2). This process was repeated 1000 times to minimize bias due to data partitioning. Based on MPC, three communication properties of the WMC, and FC, we then used the same procedures to quantify SC–FC coupling, the correlation between age and SC–FC coupling and gene weights in both S1 and S2. Finally, we assessed the consistency of results by calculating Pearson's correlation coefficients of the relationships between S1 and WD, S2 and WD, and S1 and S2.

## Acknowledgements

The authors thank all the volunteers for their participation in the study and anonymous reviewers for their insightful comments and suggestions. This work was supported by the STI2030-Major Projects (2021ZD0200500, 2022ZD0213300), National Natural Science Foundation of China (32271145, 81871425), Fundamental Research Funds for the Central Universities (2017XTCX04), and Open Research Fund of the State Key Laboratory of Cognitive Neuroscience and Learning (CNLZD2101). Data in this publication were provide (in part) by the Human Connectome Project-Development (HCP-D), which is supported by the National Institute of Mental Health of the National Institutes of Health under Award Number U01MH109589 and by funds provided by the McDonnell Center for Systems Neuroscience at Washington University in St. Louis.

## Additional information

### Funding

| Funder | Grant reference number | Author |
| --- | --- | --- |
| STI2030-Major Projects | 2021ZD0200500 | Ni Shu |
| STI2030-Major Projects | 2022ZD0213300 | Ni Shu |
| National Nature Science Foundation of China | 32271145 | Ni Shu |
| National Nature Science Foundation of China | 81871425 | Ni Shu |
| Fundamental Research Funds for the Central Universities | 2017XTCX04 | Ni Shu |
| State Key Laboratory of Cognitive Neuroscience and Learning | Open Research Fund CNLZD2101 | Ni Shu |

| Funder | Grant reference number | Author |
|---|---|---|

The funders had no role in study design, data collection, and interpretation, or the decision to submit the work for publication.

## Author contributions

Guozheng Feng, Conceptualization, Data curation, Software, Formal analysis, Validation, Visualization, Methodology, Writing - original draft, Writing - review and editing; Yiwen Wang, Weijie Huang, Haojie Chen, Jian Cheng, Data curation, Formal analysis; Ni Shu, Conceptualization, Formal analysis, Funding acquisition, Writing - review and editing

## Author ORCIDs

Guozheng Feng ⓘ http://orcid.org/0000-0002-6937-8592
Weijie Huang ⓘ http://orcid.org/0000-0002-2481-1188
Ni Shu ⓘ http://orcid.org/0000-0003-2420-2910

## Ethics

All participants or their parents (for participants under the age of 18 years) provided written informed consent and assent. The study was approved by the Institutional Review Board of Washington University in St. Louis.

Reviewer #1 (Public review): https://doi.org/10.7554/eLife.93325.3.sa1
Author response https://doi.org/10.7554/eLife.93325.3.sa2

# Additional files

## Supplementary files

• MDAR checklist

## Data availability

The HCP-D 2.0 release data that support the findings of this study are publicly available at https://www.humanconnectome.org/study/hcp-lifespan-development. R4.1.2 software (https://www.r-project.org/) was used to construct the general linear model. MATLAB scripts used for preprocessing of the AHBA dataset can be found at https://github.com/BMHLab/AHBAprocessing (*Arnatkeviciute, 2021*). Python scripts used to perform PLS regression can be found at https://scikit-learn.org/. The minimal preprocessing pipelines can be accessed at https://github.com/Washington-University/HCPpipelines (*Brown et al., 2024*). The code relevant to this study can be accessed through the following GitHub repository: https://github.com/FelixFengCN/SC-FC-coupling-development (copy archived at *Feng, 2024*).

The following previously published dataset was used:

| Author(s) | Year | Dataset title | Dataset URL | Database and Identifier |
|---|---|---|---|---|
| Somerville LH, Bookheimer SY, Buckner RL, Burgess GC, Curtiss SW, Dapretto M, Elam JS, Gaffrey MS, Harms MP, Hodge C, Kandala S, Kastman EK, Nichols TE, Schlaggar BL, Smith SM, Thomas KM, Yacoub E, Van Essen DC, Barch DM | 2021 | HCP-Development Lifespan 2.0 Release | https://www.humanconnectome.org/study/hcp-lifespan-development | NIMH Data Archive, hcp-lifespan-development |

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
