## [Editor Report · eLife assessment]

This study presents a **useful** exploration of the complex relationship between structure and function in the developing human brain using a large-scale imaging dataset from the Human Connectome Project in Development and gene expression profiles from the Allen Brain Atlas. The evidence supporting the claims of the authors is **solid**, although the inclusion of more systematic analyses of structural and functional connectivity with respect to myelin measures and oligodendrocyte-related genes, and also more details regarding the imaging analyses, cognitive scores, and design and validation strategies, would have strengthened the paper. The work will be of interest to developmental biologists and neuroscientists seeking to elucidate structure-function relationships in the human brain.

---

## [Referee Report · Reviewer #1 (Public review)]

Summary:

This work studies spatio-temporal patterns of structure-function coupling in developing brains, using a large set of imaging data acquired from children aged 5-22. Magnetic resonance imaging data of brain structure and function were obtained from a publicly available database, from which structural and functional features and measures were derived. The authors examined the spatial patterns of structure-function coupling and how they evolve with brain development. This work further sought correlations of brain structure-function coupling with behavior and explored evolutionary, microarchitectural and genetic bases that could potentially account for the observed patterns.

Strength:

The strength of this work is the use of currently available state-of-the-art analysis methods, along with a large set of high-quality imaging data, and comprehensive examinations of structure-function coupling in developing brains. The results are comprehensive and illuminating.

Weakness:

As with most other studies, transcriptomic and cellular architectures of structure-function coupling were characterized only on the basis of a common atlas in this work.

The authors have achieved their aims in this study, and the findings provide mechanistic insights into brain development, which will inspire further basic and clinical studies along this line.

---

## [Author Response]

The following is the authors’ response to the original reviews.

**Recommendations for the authors:**

**Reviewer #1 (Recommendations For The Authors):**
(1) Lines 40-42: The sentence "The coupling of structural connectome (SC) and functional connectome (FC) varies greatly across different cortical regions reflecting anatomical and functional hierarchies as well as individual differences in cognitive function, and is regulated by genes" is a misstatement. Regional variations of structure-function coupling do not really reflect differences in cognitive function among individuals, but inter-subject variations do.

Thank you for your comment. We have made revisions to the sentence to correct its misstatement. Please see lines 40-43: “The coupling of structural connectome (SC) and functional connectome (FC) varies greatly across different cortical regions reflecting anatomical and functional hierarchies[1, 6-9] and is regulated by genes[6, 8], as well as its individual differences relates to cognitive function[8, 9].”

(2) In Figure 1, the graph showing the relation between intensity and cortical depth needs explanation.

Thank you for your comment. We have added necessary explanation, please see lines 133-134: “The MPC was used to map similarity networks of intracortical microstructure (voxel intensity sampled in different cortical depth) for each cortical node.”

(3) Line 167: Change "increased" to "increase".

We have corrected it, please see lines 173-174: “…networks significantly increased with age and exhibited greater increase.”

(4) Line 195: Remove "were".

We have corrected it, please see line 204: “…default mode networks significantly contributed to the prediction…”

(5) Lines 233-240, Reproducibility analyses: Comparisons of parcellation templates were not made with respect to gene weights. Is there any particular reason?

Thank you for your comment. We have quantified the gene weights based on HCPMMP using the same procedures. We identified a correlation (*r* = 0.25, *p*<0.001) between the gene weights in HCPMMP and BNA. Given that this is a relatively weak correlation, we need to clarify the following points.

Based on HCPMMP, we produced an averaged gene expression profile for 10,027 genes covering 176 left cortical regions[1]. The excluding 4 cortical regions that had an insufficient number of assigned samples may lead to different templates having a relatively weak correlation of gene associations. Moreover, the effect of different template resolutions on the results of human connectome-transcriptome association is still unclear.

In brain connectome analysis, the choice of parcellation templates can indeed influence the subsequent findings to some extent. A methodological study[2] provided referenced correlations about 0.4~0.6 for white matter connectivity and 0.2~0.4 for white matter nodal property between two templates (refer to Figure 4 and 5 in [2]). Therefore, the age-related coupling changes as a downstream analysis was calculated using multimodal connectome and correlated with gene expression profiles, which may be influenced by the choice of templates.

We have further supplemented gene weights results obtained from HCPMMP to explicitly clarify the dependency of parcellation templates.

Please see lines 251-252: “The gene weights of HCPMMP was consistent with that of BNA (*r* = 0.25, *p* < 0.001).”

**Author response image 1. sa2fig1:** The consistency of gene weights between HCPMMP and BNA.

Please see lines 601-604: “Finally, we produced an averaged gene expression profile for 10,027 genes covering 176 left cortical regions based on HCPMMP and obtained the gene weights by PLS analysis. We performed Pearson's correlation analyses to assess the consistency of gene weights between HCPMMP and BNA.”

**Reviewer #2 (Recommendations For The Authors):**
Your paper is interesting to read and I found your efforts to evaluate the robustness of the results of different parcellation strategies and tractography methods very valuable. The work is globally easy to navigate and well written with informative good-quality figures, although I think some additional clarifications will be useful to improve readability. My suggestions and questions are detailed below (I aimed to group them by topic which did not always succeed so apologies if the comments are difficult to navigate, but I hope they will be useful for reflection and to incorporate in your work).* L34: 'developmental disorder'** As far as I understand, the subjects in HCP-D are mostly healthy (L87). Thus, while your study provides interesting insights into typical brain development, I wonder if references to 'disorder' might be premature. In the future, it would be interesting to extend your approach to the atypical populations. In any case, it would be extremely helpful and appreciated if you included a figure visualising the distribution of behavioural scores within your population and in relationship to age at scan for your subjects (and to include a more detailed description of the assessment in the methods section) given that large part of your paper focuses on their prediction using coupling inputs (especially given a large drop of predictive performance after age correction). Such figures would allow the reader to better understand the cognitive variability within your data, but also potential age relationships, and generally give a better overview of your cohort.

We agree with your comment that references to 'disorder' is premature. We have made revisions in abstract and conclusion.

Please see lines 33-34: “This study offers insight into the maturational principles of SC-FC coupling in typical development.”

Please see lines 395-396: “Further investigations are needed to fully explore the clinical implications of SC-FC coupling for a range of developmental disorders.”

In addition, we have included a more detailed description of the cognitive scores in the methods section and provided a figure to visualize the distributions of cognitive scores and in relationship to age for subjects. Please see lines 407-413: “Cognitive scores. We included 11 cognitive scores which were assessed with the National Institutes of Health (NIH) Toolbox Cognition Battery (https://www.healthmeasures.net/exploremeasurement-systems/nih-toolbox), including episodic memory, executive function/cognitive flexibility, executive function/inhibition, language/reading decoding, processing speed, language/vocabulary comprehension, working memory, fluid intelligence composite score, crystal intelligence composite score, early child intelligence composite score and total intelligence composite score. Distributions of these cognitive scores and their relationship with age are illustrated in Figure S12.”

**Author response image 2. sa2fig2:** Cognitive scores and age distributions of scans.

* SC-FC coupling** L162: 'Regarding functional subnetworks, SC-FC coupling increased disproportionately with age (Figure 3C)'.*** As far as I understand, in Figure 3C, the points are the correlation with age for a given ROI within the subnetwork. Is this correct? If yes, I am not sure how this shows a disproportionate increase in coupling. It seems that there is great variability of SC-FC correlation with age across regions within subnetworks, more so than the differences between networks. This would suggest that the coupling with age is regionally dependent rather than network-dependent? Maybe you could clarify?

The points are the correlation with age for a given ROI within the subnetwork in Figure 3C. We have revised the description, please see lines 168-174: “Age correlation coefficients distributed within functional subnetworks were shown in Figure 3C. Regarding mean SC-FC coupling within functional subnetworks, the somatomotor (𝛽𝑎𝑔𝑒=2.39E-03, F=4.73, *p*=3.10E-06, *r*=0.25, *p*=1.67E07, Figure 3E), dorsal attention (𝛽𝑎𝑔𝑒=1.40E-03, F=4.63, *p*=4.86E-06, *r*=0.24, *p*=2.91E-07, Figure 3F), frontoparietal (𝛽𝑎𝑔𝑒 = 2.11E-03, F=6.46, *p*=2.80E-10, *r*=0.33, *p*=1.64E-12, Figure 3I) and default mode (𝛽𝑎𝑔𝑒 = 9.71E-04, F=2.90, *p*=3.94E-03, *r*=0.15, *p*=1.19E-03, Figure 3J) networks significantly increased with age and exhibited greater increase.” In addition, we agree with your comment that the coupling with age is more likely region-dependent than network-dependent. We have added the description, please see lines 329-332: “We also found the SC-FC coupling with age across regions within subnetworks has more variability than the differences between networks, suggesting that the coupling with age is more likely region-dependent than network-dependent.” This is why our subsequent analysis focused on regional coupling.

*** Additionally, we see from Figure 3C that regions within networks have very different changes with age. Given this variability (especially in the subnetworks where you show both positive and negative correlations with age for specific ROIs (i.e. all of them)), does it make sense then to show mean coupling over regions within the subnetworks which erases the differences in coupling with age relationships across regions (Figures 3D-J)?

Considering the interest and interpretation for SC-FC coupling, showing the mean coupling at subnetwork scales with age correlation is needed, although this eliminates variability at regional scale. These results at different scales confirmed that coupling changes with age at this age group are mainly increased.

*** Also, I think it would be interesting to show correlation coefficients across all regions, not only the significant ones (3B). Is there a spatially related tendency of increases/decreases (rather than a 'network' relationship)? Would it be interesting to show a similar figure to Figure S7 instead of only the significant regions?

As your comment, we have supplemented the graph which shows correlation coefficients across all regions into Figure 3B. Similarly, we supplemented to the other figures (Figure S3-S6).

**Author response image 3. sa2fig3:** Aged-related changes in SC-FC coupling. (A) Increases in whole-brain coupling with age. (B) Correlation of age with SC-FC coupling across all regions and significant regions (*p*<0.05, FDR corrected). (C) Comparisons of age-related changes in SC-FC coupling among functional networks. The boxes show the median and interquartile range (IQR; 25–75%), and the whiskers depict 1.5× IQR from the first or third quartile. (D-J) Correlation of age with SC-FC coupling across the VIS, SM, DA, VA, LIM, FP and DM. VIS, visual network; SM, somatomotor network; DA, dorsal attention network; VA, ventral attention network; LIM, limbic network; FP, frontoparietal network; DM, default mode network.

*** For the quantification of MPC.**** L421: you reconstructed 14 cortical surfaces from the wm to pial surface. If we take the max thickness of the cortex to be 4.5mm (Fischl & Dale, 2000), the sampling is above the resolution of your anatomical images (0.8mm). Could you expand on what the interest is in sampling such a higher number of surfaces given that the resolution is not enough to provide additional information?

The surface reconstruction was based on state-of-the-art equivolumetric surface construction techniques[3] which provides a simplified recapitulation of cellular changes across the putative laminar structure of the cortex. By referencing a 100-μm resolution Merkerstained 3D histological reconstruction of an entire post mortem human brain (BigBrain: https://bigbrain.loris.ca/main.php), a methodological study[4] systematically evaluated MPC stability with four to 30 intracortical surfaces when the resolution of anatomical image was 0.7 mm, and selected 14 surfaces as the most stable solution. Importantly, it has been proved the *in vivo* approach can serve as a lower resolution yet biologically meaningful extension of the histological work[4].

**** L424: did you aggregate intensities over regions using mean/median or other statistics?

It might be useful to specify.

Thank you for your careful comment. We have revised the description in lines 446-447: “We averaged the intensity profiles of vertices over 210 cortical regions according to the BNA”.

**** L426: personal curiosity, why did you decide to remove the negative correlation of the intensity profiles from the MPC? Although this is a common practice in functional analyses (where the interpretation of negatives is debated), within the context of cortical correlations, the negative values might be interesting and informative on the level of microstructural relationships across regions (if you want to remove negative signs it might be worth taking their absolute values instead).

We agree with your comment that the interpretation of negative correlation is debated in MPC. Considering that MPC is a nascent approach to network modeling, we adopted a more conservative strategy that removing negative correlation by referring to the study [4] that proposed the approach. As your comment, the negative correlation might be informative. We will also continue to explore the intrinsic information on the negative correlation reflecting microstructural relationships.

**** L465: could you please expand on the notion of self-connections, it is not completely evident what this refers to.

We have revised the description in lines 493-494: “𝑁𝑐 is the number of connection (𝑁𝑐 = 245 for BNA)”.

**** Paragraph starting on L467: did you evaluate the multicollinearities between communication models? It is possibly rather high (especially for the same models with similar parameters (listed on L440-444)). Such dependence between variables might affect the estimates of feature importance (given the predictive models only care to minimize error, highly correlated features can be selected as a strong predictor while the impact of other features with similarly strong relationships with the target is minimized thus impacting the identification of reliable 'predictors').

We agree with your comment. The covariance structure (multicollinearities) among the communication models have a high probability to lead to unreliable predictor weights. In our study, we applied Haufe's inversion transform[5] which resolves this issue by computing the covariance between the predicted FC and each communication models in the training set. More details for Haufe's inversion transform please see [5]. We further clarified in the manuscript, please see in lines 497-499: “And covariance structure among the predictors may lead to unreliable predictor weights. Thus, we applied Haufe's inversion transform[38] to address these issues and identify reliable communication mechanisms.”

**** L474: I am not completely familiar with spin tests but to my understanding, this is a spatial permutation test. I am not sure how this applies to the evaluation of the robustness of feature weight estimates per region (if this was performed per region), it would be useful to provide a bit more detail to make it clearer.

As your comment, we have supplemented the detail, please see lines 503-507: “Next, we generated 1,000 FC permutations through a spin test[86] for each nodal prediction in each subject and obtained random distributions of model weights. These weights were averaged over the group and were investigated the enrichment of the highest weights per region to assess whether the number of highest weights across communication models was significantly larger than that in a random discovery.”

**** L477: 'significant communication models were used to represent WMC...', but in L103 you mention you select 3 models: communicability, mean first passage, and flow graphs. Do you want to say that only 3 models were 'significant' and these were exactly the same across all regions (and data splits/ parcellation strategies/ tractography methods)? In the methods, you describe a lot of analysis and testing but it is not completely clear how you come to the selection of the final 3, it would be beneficial to clarify. Also, the final 3 were selected on the whole dataset first and then the pipeline of SC-FC coupling/age assessment/behaviour predictions was run for every (WD, S1, S2) for both parcellations schemes and tractography methods or did you end up with different sets each time? It would be good to make the pipeline and design choices, including the validation bit clearer (a figure detailing all the steps which extend Figure 1 would be very useful to understand the design/choices and how they relate to different runs of the validation).

Thank you for your comment. In all reproducibility analyses, we used the same 3 models which was selected on the main pipeline (probabilistic tractography and BNA parcellation). According to your comment, we produced a figure that included the pipeline of model selection as the extend of Figure 1. And the description please see lines 106-108: “We used these three models to represent the extracortical connectivity properties in subsequent discovery and reproducibility analyses (Figure S1).”

**Author response image 4. sa2fig4:** Pipeline of model selection and reproducibility analyses.

**** Might the imbalance of features between structural connectivity and MPC affect the revealed SC-FC relationships (3 vs 1)? Why did you decide on this ratio rather than for example best WM structural descriptor + MPC?

We understand your concern. The WMC communication models represent diverse geometric, topological, or dynamic factors. In order to describe the properties of WMC as best as possible, we selected three communication models after controlling covariance structure that can significantly predict FC from the 27 models. Compared to MPC, this does present a potential feature imbalance problem. However, this still supports the conclusion that coupling models that incorporate microarchitectural properties yield more accurate predictions of FC from SC[6, 7]. The relevant experiments are shown in Figure S2 below. If only the best WM structural descriptor is used, this may lose some communication properties of WMC.

**** L515: were intracranial volume and in-scanner head motion related to behavioural measures? These variables likely impact the inputs, do you expect them to influence the outcome assessments? Or is there a mistake on L518 and you actually corrected the input features rather than the behaviour measures?

The in-scanner head motion and intracranial volume are related to some age-adjusted behavioural measures, as shown in the following table. The process of regression of covariates from cognitive measures was based on these two cognitive prediction studies [8, 9]. Please see lines 549-554: “Prior to applying the nested fivefold cross-validation framework to each behaviour measure, we regressed out covariates including sex, intracranial volume, and in-scanner head motion from the behaviour measure[59, 69]. Specifically, we estimated the regression coefficients of the covariates using the training set and applied them to the testing set. This regression procedure was repeated for each fold.”

**Author response table 1. sa2table1:** 

	In-scanner head motion		Intracranial volume	
	r value	p value	r value	p value
Episodic memory	-0.04	0.494825	0.00	0.963828
Executive function/cognitiveflexibility	-0.13	0.017108	-0.02	0.729339
Executive function/inhibition,language/reading decoding	0.00	0.985376	-0.01	0.911467
Language/reading decoding	-0.09	0.124475	0.03	0.596947
Processing speed	-0.07	0.215465	0.11	0.049091
Language/vocabularycomprehension	-0.15	0.007839	0.01	0.835508
Working memory	-0.02	0.767743	0.08	0.16235
Fluid intelligence composite scoreCrystal intelligence compositescore	-0.12	0.034428	0.02	0.695776
Early child intelligence composite	-0.09	0.112609	0.08	0.153797
score	-0.09	0.107082	0.04	0.503285
Total intelligence composite score	-0.12	0.037287	0.07	0.219497
Age-adjusted cognitive score				

** Additionally, in the paper, you propose that the incorporation of cortical microstructural (myelin-related) descriptors with white-matter connectivity to explain FC provides for 'a more comprehensive perspective for characterizing the development of SC-FC coupling' (L60). This combination of cortical and white-matter structure is indeed interesting, however the benefits of incorporating different descriptors could be studied further. For example, comparing results of using only the white matter connectivity (assessed through selected communication models) ~ FC vs (white matter + MPC) ~ FC vs MPC ~ FC. Which descriptors better explain FC? Are the 'coupling trends' similar (or the same)? If yes, what is the additional benefit of using the more complex combination? This would also add strength to your statement at L317: 'These discrepancies likely arise from differences in coupling methods, highlighting the complementarity of our methods with existing findings'. Yes, discrepancies might be explained by the use of different SC inputs. However, it is difficult to see how discrepancies highlight complementarity - does MCP (and combination with wm) provide additional information to using wm structural alone?~

According to your comment, we have added the analyses based on different models using only the myelin-related predictor or WM connectivity to predict FC, and further compared the results among different models. please see lines 519-521: “In addition, we have constructed the models using only MPC or SCs to predict FC, respectively. Spearman’s correlation was used to assess the consistency between spatial patterns based on different models.”

Please see lines 128-130: “In addition, the coupling pattern based on other models (using only MPC or only SCs to predict FC) and the comparison between the models were shown in Figure S2A-C.” Please see lines 178-179: “The age-related patterns of SC-FC coupling based other coupling models were shown in Figure S2D-F.”

Although we found that there were spatial consistencies in the coupling patterns between different models, the incorporation of MPC with SC connectivity can improve the prediction of FC than the models based on only MPC or SC. For age-related changes in coupling, the differences between the models was further amplified. We agree with you that the complementarity cannot be explicitly quantified and we have revised the description, please see line 329: “These discrepancies likely arise from differences in coupling methods.”

**Author response image 5. sa2fig5:** Comparison results between different models. Spatial pattern of mean SC-FC coupling based on MPC ~ FC (A), SCs ~ FC (B), and MPC + SCs ~ FC (C). Correlation of age with SC-FC coupling across cortex based on MPC ~ FC (D), SCs ~ FC (E), and MPC + SCs ~ FC (F).

** For the interpretation of results: L31 'SC-FC coupling is positively associated with genes in oligodendrocyte-related pathways and negatively associated with astrocyte-related gene'; L124: positive myelin content with SC-FC coupling...and similarly on L81, L219, L299, L342, and L490:***You use a T1/T2 ratio which is (in large part) a measure of myelin to estimate the coupling between SC and FC. Evaluation with SC-FC coupling with myeline described in Figure 2E is possibly biased by the choice of this feature. Similarly, it is possible that reported positive associations with oligodendrocyte-related pathways and SC-FC coupling in your work could in part result from a bias introduced by the 'myelin descriptor' (conversely, picking up the oligodendrocyte-related genes is a nice corroboration for the T1/T2 ration being a myelin descriptor, so that's nice). However, it is possible that if you used a different descriptor of the cortical microstructure, you might find different expression patterns associated with the SCFC coupling (for example using neurite density index might pick up neuronal-related genes?). As mentioned in my previous suggestions, I think it would be of interest to first use only the white matter structural connectivity feature to assess coupling to FC and assess the gene expression in the cortical regions to see if the same genes are related, and subsequently incorporate MPC to dissociate potential bias of using a myelin measure from genetic findings.

Thank you for your insightful comments. In this paper, however, the core method of measuring coupling is to predict functional connections using multimodal structural connections, which may yield more information than a single modal. We agree with your comment that separating SCs and MPC to look at the genes involved in both separately could lead to interesting discoveries. We will continue to explore this in the future.

** Generally, I find it difficult to understand the interpretation of SC-FC coupling measures and would be interested to hear your thinking about this. As you mention on L290-294, how well SC predicts FC depends on which input features are used for the coupling assessment (more complex communication models, incorporating additional microstructural information etc 'yield more accurate predictions of FC' L291) - thus, calculated coupling can be interpreted as a measure of how well a particular set of input features explain FC (different sets will explain FC more or less well) ~ coupling is related to a measure of 'missing' information on the SC-FC relationship which is not contained within the particular set of structural descriptors - with this approach, the goal might be to determine the set that best, i.e. completely, explains FC to understand the link between structure and function. When you use the coupling measures for comparisons with age, cognition prediction etc, the 'status' of the SC-FC changes, it is no longer the amount of FC explained by the given SC descriptor set, but it's considered a descriptor in itself (rather than an effect of feature selection / SC-FC information overlap) - how do you interpret/argue for this shift of use?

Thank you for your comment. In this paper, we obtain reasonable SC-FC coupling by determining the optimal set of structural features to explain the function. The coupling essentially measures the direct correspondence between structure and function. To study the relationship between coupling and age and cognition is actually to study the age correlation and cognitive correlation of this direct correspondence between structure and function.

** In a similar vein to the above comment, I am interested to hear what you think: on L305 you mention that 'perfect SC-FC coupling may be unlikely'. Would this reasoning suggest that functional activity takes place through other means than (and is therefore somehow independent of) biological (structural) substrates? For now, I think one can only say that we have imperfect descriptors of the structure so there is always information missing to explain function, this however does not mean the SC and FC are not perfectly coupled (only that we look at insufficient structural descriptors - limitations of what imaging can assess, what we measure etc). This is in line with L305 where you mention that 'Moreover, our results suggested that regional preferential contributions across different SCs lead to variations in the underlying communication process'. This suggests that locally different areas might use different communication models which are not reflected in the measures of SC-FC coupling that was employed, not that the 'coupling' is lower or higher (or coupling is not perfect). This is also a change in approach to L293: 'This configuration effectively releases the association cortex from strong structural constraints' - the 'release' might only be in light of the particular structural descriptors you use - is it conceivable that a different communication model would be more appropriate (and show high coupling) in these areas.

Thank you for your insightful comments. We have changed the description, please see lines 315317: “SC-FC coupling is dynamic and changes throughout the lifespan[7], particularly during adolescence[6,9], suggesting that perfect SC-FC coupling may require sufficient structural descriptors.”

*Cognitive predictions:** From a practical stand-point, do you think SC-FC coupling is a better (more accurate) indicator of cognitive outcomes (for example for future prediction studies) than each modality alone (which is practically easier to obtain and process)? It would be useful to check the behavioural outcome predictions for each modality separately (as suggested above for coupling estimates). In case SC-FC coupling does not outperform each modality separately, what is the benefit of using their coupling? Similarly, it would be useful to compare to using only cortical myelin for the prediction (which you showed to increase in importance for the coupling). In the case of myelin->coupling-> intelligence, if you are able to predict outcomes with the same performance from myelin without the need for coupling measures, what is the benefit of coupling?

From a predictive performance point of view, we do not believe that SC-FC coupling is a better indicator than a single mode (voxel, network or other indicator). Our starting point is to assess whether SC-FC coupling is related to the individual differences of cognitive performances rather than to prove its predictive power over other measures. As you suggest, it's a very interesting perspective on the predictive power of cognition by separating the various modalities and comparing them. We will continue to explore this issue in the future study.

** The statement on L187 'suggesting that increased SC-FC coupling during development is associated with higher intelligence' might not be completely appropriate before age corrections (especially given the large drop in performance that suggests confounding effects of age).

According to your comment, we have removed the statement.

** L188: it might be useful to report the range of R across the outer cross-validation folds as from Figure 4A it is not completely clear that the predictive performance is above the random (0) threshold. (For the sake of clarity, on L180 it might be useful for the reader if you directly report that other outcomes were not above the random threshold).

According to your comment, we have added the range of R and revised the description, please see lines 195-198: “Furthermore, even after controlling for age, SC-FC coupling remained a significant predictor of general intelligence better than at chance (Pearson’s *r*=0.11±0.04, *p*=0.01, FDR corrected, Figure 4A). For fluid intelligence and crystal intelligence, the predictive performances of SC-FC coupling were not better than at chance (Figure 4A).”

In a similar vein, in the text, you report Pearson's R for the predictive results but Figure 4A shows predictive accuracy - accuracy is a different (categorical) metric. It would be good to homogenise to clarify predictive results.

We have made the corresponding changes in Figure 4.

**Author response image 6. sa2fig6:** Encoding individual differences in intelligence using regional SC-FC coupling. (A) Predictive accuracy of fluid, crystallized, and general intelligence composite scores. (B) Regional distribution of predictive weight. (C) Predictive contribution of functional networks. The boxes show the median and interquartile range (IQR; 25–75%), and the whiskers depict the 1.5× IQR from the first or third quartile.

*Methods and QC:-Parcellations** It would be useful to mention briefly how the BNA was applied to the data and if any quality checks were performed for the resulting parcellations, especially for the youngest subjects which might be most dissimilar to the population used to derive the atlas (healthy adults HCP subjects) ~ question of parcellation quality.

We have added the description, please see lines 434-436: “The BNA[31] was projected on native space according to the official scripts (http://www.brainnetome.org/resource/) and the native BNA was checked by visual inspection.”

** Additionally, the appropriateness of structurally defined regions for the functional analysis is also a topic of important debate. It might be useful to mention the above as limitations (which apply to most studies with similar focus).

We have added your comment to the methodological issues, please see lines 378-379: “Third, the appropriateness of structurally defined regions for the functional analysis is also a topic of important debate.”

- Tractography** L432: it might be useful to name the method you used (probtrackx).

We have added this name to the description, please see lines 455-456: “probabilistic tractography (probtrackx)[78, 79] was implemented in the FDT toolbox …”

** L434: 'dividing the total fibres number in source region' - dividing by what?

We have revised the description, please see line 458: “dividing by the total fibres number in source region.”

** L436: 'connections in subcortical areas were removed' - why did you trace connections to subcortical areas in the first place if you then removed them (to match with cortical MPC areas I suspect)? Or do you mean there were spurious streamlines through subcortical regions that you filtered?

On the one hand we need to match the MPC, and on the other hand, as we stated in methodological issues, the challenge of accurately resolving the connections of small structures within subcortical regions using whole-brain diffusion imaging and tractography techniques[10, 11].

** Following on the above, did you use any exclusion masks during the tracing? In general, more information about quality checks for the tractography would be useful. For example, L437: did you do any quality evaluations based on the removed spurious streamlines? For example, were there any trends between spurious streamlines and the age of the subject? Distance between regions/size of the regions?

We did not use any exclusion masks. We performed visual inspection for the tractography quality and did not assess the relationship between spurious streamlines and age or distance between regions/size of the regions.

** L439: 'weighted probabilistic network' - this was weighted by the filtered connectivity densities or something else?

The probabilistic network is weighted by the filtered connectivity densities.

** I appreciate the short description of the communication models in Text S1, it is very useful.

Thank you for your comment.

** In addition to limitations mentioned in L368 - during reconstruction, have you noticed problems resolving short inter-hemispheric connections?

We have not considered this issue, we have added it to the limitation, please see lines 383-384: “In addition, the reconstruction of short connections between hemispheres is a notable challenge.”

- Functional analysis:** There is a difference in acquisition times between participants below and above 8 years (21 vs 26 min), does the different length of acquisition affect the quality of the processed data?

We have made relatively strict quality control to ensure the quality of the processed data.

** L446 'regressed out nuisance variables' - it would be informative to describe in more detail what you used to perform this.

We have provided more detail about the regression of nuisance variables, please see lines 476-477: “The nuisance variables were removed from time series based on general linear model.”

** L450-452: it would be useful to add the number of excluded participants to get an intuition for the overall quality of the functional data. Have you checked if the quality is associated with the age of the participant (which might be related to motion etc). Adding a distribution of remaining frames across participants (vs age) would be useful to see in the supplementary methods to better understand the data you are using.

We have supplemented the exclusion information of the subjects during the data processing, and the distribution and aged correlation of motion and remaining frames. Please see lines 481-485: “Quality control. The exclusion of participants in the whole multimodal data processing pipeline was depicted in Figure S13. In the context of fMRI data, we computed Pearson’s correlation between motion and age, as well as between the number of remaining frames and age, for the included participants aged 5 to 22 years and 8 to 22 years, respectively. These correlations were presented in Figure S14.”

**Author response image 7. sa2fig7:** Exclusion of participants in the whole multimodal data processing pipeline.

**Author response image 8. sa2fig8:** Figure S14. Correlations between motion and age and number of remaining frames and age.

** L454: 'Pearson's correlation's... ' In contrast to MPC you did not remove negative correlations in the functional matrices. Why this choice?

Whether the negative correlation connection of functional signal is removed or not has always been a controversial issue. Referring to previous studies of SC-FC coupling[12-14], we find that the practice of retaining negative correlation connections has been widely used. In order to retain more information, we chose this strategy. Considering that MPC is a nascent approach to network modeling, we adopted a more conservative strategy that removing negative correlation by referring to the study [4] that proposed the approach.

- Gene expression:** L635, you focus on the left cortex, is this common? Do you expect the gene expression to be fully symmetric (given reported functional hemispheric asymmetries)? It might be good to expand on the reasoning.

An important consideration regarding sample assignment arises from the fact that only two out of six brains were sampled from both hemispheres and four brains have samples collected only in the left. This sparse sampling should be carefully considered when combining data across donors[1]. We have supplemented the description, please see lines 569-571: “Restricting analyses to the left hemisphere will minimize variability across regions (and hemispheres) in terms of the number of samples available[40].”

** Paragraph of L537: you use evolution of coupling with age (correlation) and compare to gene expression with adults (cohort of Allen Human Brain Atlas - no temporal evolution to the gene expressions) and on L369 you mention that 'relative spatial patterns of gene expressions remain stable after birth'. Of course this is not a place to question previous studies, but would you really expect the gene expression associated with the temporary processes to remain stable throughout the development? For example, myelination would follow different spatiotemporal gradient across brain regions, is it reasonable to expect that the expression patterns remain the same? How do you then interpret a changing measure of coupling (correlation with age) with a gene expression assessed statically?

We agree with your comment that the spatial expression patterns is expected to vary at different periods. We have revised the previous description, please see lines 383-386: “Fifth, it is important to acknowledge that changes in gene expression levels during development may introduce bias in the results.”

- Reproducibility analyses:** Paragraph L576: are we to understand that you performed the entire pipeline 3 times (WD, S1, S2) for both parcellations schemes and tractography methods (~12 times) including the selection of communication models and you always got the same best three communication models and gene expression etc? Or did you make some design choices (i.e. selection of communication models) only on a specific set-up and transfer to other settings?

The choice of communication model is established at the beginning, which we have clarified in the article, please see lines 106-108: “We used these three models to represent the extracortical connectivity properties in subsequent discovery and reproducibility analyses (Figure S1).” For reproducibility analyses (parcellation, tractography, and split-half validation), we fixed other settings and only assessed the impact of a single factor.

** Paragraph of L241: I really appreciate you evaluated the robustness of your results to different tractography strategies. It is reassuring to see the similarity in results for the two approaches. Did you notice any age-related effects on tractography quality for the two methods given the wide age range (did you check?)

In our study, the tractography quality was checked by visual inspection. Using quantifiable tools to tractography quality in future studies could answer this question objectively.

** Additionally, I wonder how much of that overlap is driven by the changes in MPC which is the same between the two methods... especially given its high weight in the SC-FC coupling you reported earlier in the paper. It might be informative to directly compare the connectivity matrices derived from the two tracto methods directly. Generally, as mentioned in the previous comments, I think it would be interesting to assess coupling using different input settings (with WM structural and MPC separate and then combined).

As your previous comment, we have examined the coupling patterns, coupling differences, coupling age correlation, and spatial correlations between the patterns based on different models, as shown in Figure S2. Please see our response to the previous comment for details.

** L251 - I also wonder if the random splitting is best adapted to validation in your case given you study relationships with age. Would it make more sense to make stratified splits to ensure a 'similar age coverage' across splits?

In our study, we adopt the random splitting process which repeated 1,000 times to minimize bias due to data partitioning. The stratification you mentioned is a reasonable method, and keeping the age distribution even will lead to higher verification similarity than our validation method. However, from the validation results of our method, the similarity is sufficient to explain the generalization of our findings.

Minor commentsL42: 'is regulated by genes'** Coupling (if having a functional role and being regulated at all) is possibly resulting from a complex interplay of different factors in addition to genes, for example, learning/environment, it might be more cautious to use 'regulated in part by genes' or similar.

We have corrected it, please see line 42.

L43 (and also L377): 'development of SC-FC coupling'** I know this is very nitpicky and depends on your opinion about the nature of SC-FC coupling, but 'development of SC-FC coupling' gives an impression of something maturing that has a role 'in itself' (for example development of eye from neuroepithelium to mature organ etc.). For now, I am not sure it is fully certain that SC-FC coupling is more than a byproduct of the comparison between SC and FC, using 'changes in SC-FC coupling with development' might be more apt.

We have corrected it, please see lines 43-44.

L261 'SC-FC coupling was stronger ... [] ... and followed fundamental properties of cortical organization.' vs L168 'No significant correlations were found between developmental changes in SC-FC coupling and the fundamental properties of cortical organization'.**Which one is it? I think in the first you refer to mean coupling over all infants and in the second about correlation with age. How do you interpret the difference?

Between the ages of 5 and 22 years, we found that the mean SC-FC coupling pattern has become similar to that of adults, consistent with the fundamental properties of cortical organization. However, the developmental changes in SC-FC coupling are heterogeneous and sequential and do not follow the mean coupling pattern to change in the same magnitude.

L277: 'temporal and spatial complexity'** Additionally, communication models have different assumptions about the flow within the structural network and will have different biological plausibility (they will be more or less 'realistic').

Here temporal and spatial complexity is from a computational point of view.

L283: 'We excluded a centralized model (shortest paths), which was not biologically plausible' ** But in Text S1 and Table S1 you specify the shortest paths models. Does this mean you computed them but did not incorporate them in the final coupling computations even if they were predictive?** Generally, I find the selection of the final 3 communication models confusing. It would be very useful if you could clarify this further, for example in the methods section.

We used all twenty-seven communication models (including shortest paths) to predict FC at the node level for each participant. Then we identified three communication models that can significantly predict FC. For the shortest path, he was excluded because he did not meet the significance criteria. We have further added methodological details to this section, please see lines 503-507.

L332 'As we observed increasing coupling in these [frontoparietal network and default mode network] networks, this may have contributed to the improvements in general intelligence, highlighting the flexible and integrated role of these networks' vs L293 'SC-FC coupling in association areas, which have lower structural connectivity, was lower than that in sensory areas. This configuration effectively releases the association cortex from strong structural constraints imposed by early activity cascades, promoting higher cognitive functions that transcend simple sensori-motor exchanges'** I am not sure I follow the reasoning. Could you expand on why it would be the decoupling promoting the cognitive function in one case (association areas generally), but on the reverse the increased coupling in frontoparietal promoting the cognition in the other (specifically frontoparietal)?

We tried to explain the problem, for general intelligence, increased coupling in frontoparietal could allow more effective information integration enable efficient collaboration between different cognitive processes.

* Formatting errors etc.L52: maybe rephrase?

We have rephrased, please see lines 51-53: “The T1- to T2-weighted (T1w/T2w) ratio of MRI has been proposed as a means of quantifying microstructure profile covariance (MPC), which reflects a simplified recapitulation in cellular changes across intracortical laminar structure[6, 1215].”

L68: specialization1,[20].

We have corrected it.

L167: 'networks significantly increased with age and exhibited greater increased' - needs rephrasing.

We have corrected it.

L194: 'networks were significantly predicted the general intelligence' - needs rephrasing.

We have corrected it, please see lines 204-205: “we found that the weights of frontoparietal and default mode networks significantly contributed to the prediction of the general intelligence.”

L447: 'and temporal bandpass filtering' - there is a verb missing.

We have corrected it, please see line 471: “executed temporal bandpass filtering.”

L448: 'greater than 0.15' - unit missing.

We have corrected it, please see line 472: “greater than 0.15 mm”.

L452: 'After censoring, regression of nuisance variables, and temporal bandpass filtering,' - no need to repeat the steps as you mentioned them 3 sentences earlier.

We have removed it.

L458-459: sorry I find this description slightly confusing. What do you mean by 'modal'? Connectional -> connectivity profile. The whole thing could be simplified, if I understand correctly your vector of independent variables is a set of wm and microstructural 'connectivity' of the given node... if this is not the case, please make it clearer.

We have corrected it, please see line 488: “where 𝒔𝑖 is the 𝑖th SC profiles, 𝑛 is the number of SC profiles”.

L479: 'values and system-specific of 480 coupling'.

We have corrected it.

L500: 'regular' - regularisation.

We have changed it to “regularization”.

L567: Do you mean that in contrast to probabilistic with FSL you use deterministic methods within Camino? For L570, you introduce communication models through 'such as': did you fit all models like before? If not, it might be clearer to just list the ones you estimated rather than introduce through 'such as'.

We have changed the description to avoid ambiguity, please see lines 608-609: “We then calculated the communication properties of the WMC including communicability, mean first passage times of random walkers, and flow graphs (timescales=1).”

Citation [12], it is unusual to include competing interests in the citation, moreover, Dr. Bullmore mentioned is not in the authors' list - this is most likely an error with citation import, it would be good to double-check.

We have corrected it.

L590: Python scripts used to perform PLS regression can 591 be found at https://scikitlearn.org/. The link leads to general documentation for sklearn.

We have corrected it, please see lines 627-630: “Python scripts used to perform PLS regression can be found at https://scikit-learn.org/stable/modules/generated/sklearn.cross_decomposition.PLSRegression.html#sklearn.cro ss_decomposition.PLSRegression.”

P26 and 27 - there are two related sections: Data and code availability and Code availability - it might be worth merging into one section if possible.

We have corrected it, please see lines 623-633.

References

(1) Arnatkeviciute A, Fulcher BD, Fornito A. A practical guide to linking brain-wide gene expression and neuroimaging data. Neuroimage. 2019;189:353-67. Epub 2019/01/17. doi: 10.1016/j.neuroimage.2019.01.011. PubMed PMID: 30648605.

(2) Zhong S, He Y, Gong G. Convergence and divergence across construction methods for human brain white matter networks: an assessment based on individual differences. Hum Brain Mapp. 2015;36(5):1995-2013. Epub 2015/02/03. doi: 10.1002/hbm.22751. PubMed PMID: 25641208; PubMed Central PMCID: PMCPMC6869604.

(3) Waehnert MD, Dinse J, Weiss M, Streicher MN, Waehnert P, Geyer S, et al. Anatomically motivated modeling of cortical laminae. Neuroimage. 2014;93 Pt 2:210-20. Epub 2013/04/23. doi: 10.1016/j.neuroimage.2013.03.078. PubMed PMID: 23603284.

(4) Paquola C, Vos De Wael R, Wagstyl K, Bethlehem RAI, Hong SJ, Seidlitz J, et al. Microstructural and functional gradients are increasingly dissociated in transmodal cortices. PLoS Biol. 2019;17(5):e3000284. Epub 2019/05/21. doi: 10.1371/journal.pbio.3000284. PubMed PMID: 31107870.

(5) Haufe S, Meinecke F, Gorgen K, Dahne S, Haynes JD, Blankertz B, et al. On the interpretation of weight vectors of linear models in multivariate neuroimaging. Neuroimage. 2014;87:96-110. Epub 2013/11/19. doi: 10.1016/j.neuroimage.2013.10.067. PubMed PMID: 24239590.

(6) Demirtas M, Burt JB, Helmer M, Ji JL, Adkinson BD, Glasser MF, et al. Hierarchical Heterogeneity across Human Cortex Shapes Large-Scale Neural Dynamics. Neuron. 2019;101(6):1181-94 e13. Epub 2019/02/13. doi: 10.1016/j.neuron.2019.01.017. PubMed PMID: 30744986; PubMed Central PMCID: PMCPMC6447428.

(7) Deco G, Kringelbach ML, Arnatkeviciute A, Oldham S, Sabaroedin K, Rogasch NC, et al. Dynamical consequences of regional heterogeneity in the brain's transcriptional landscape. Sci Adv. 2021;7(29). Epub 2021/07/16. doi: 10.1126/sciadv.abf4752. PubMed PMID: 34261652; PubMed Central PMCID: PMCPMC8279501.

(8) Chen J, Tam A, Kebets V, Orban C, Ooi LQR, Asplund CL, et al. Shared and unique brain network features predict cognitive, personality, and mental health scores in the ABCD study. Nat Commun. 2022;13(1):2217. Epub 2022/04/27. doi: 10.1038/s41467-022-29766-8. PubMed PMID: 35468875; PubMed Central PMCID: PMCPMC9038754.

(9) Li J, Bzdok D, Chen J, Tam A, Ooi LQR, Holmes AJ, et al. Cross-ethnicity/race generalization failure of behavioral prediction from resting-state functional connectivity. Sci Adv. 2022;8(11):eabj1812. Epub 2022/03/17. doi: 10.1126/sciadv.abj1812. PubMed PMID: 35294251; PubMed Central PMCID: PMCPMC8926333.

(10) Thomas C, Ye FQ, Irfanoglu MO, Modi P, Saleem KS, Leopold DA, et al. Anatomical accuracy of brain connections derived from diffusion MRI tractography is inherently limited. Proc Natl Acad Sci U S A. 2014;111(46):16574-9. Epub 2014/11/05. doi: 10.1073/pnas.1405672111. PubMed PMID: 25368179; PubMed Central PMCID: PMCPMC4246325.

(11) Reveley C, Seth AK, Pierpaoli C, Silva AC, Yu D, Saunders RC, et al. Superficial white matter fiber systems impede detection of long-range cortical connections in diffusion MR tractography. Proc Natl Acad Sci U S A. 2015;112(21):E2820-8. Epub 2015/05/13. doi: 10.1073/pnas.1418198112. PubMed PMID: 25964365; PubMed Central PMCID: PMCPMC4450402.

(12) Gu Z, Jamison KW, Sabuncu MR, Kuceyeski A. Heritability and interindividual variability of regional structure-function coupling. Nat Commun. 2021;12(1):4894. Epub 2021/08/14. doi: 10.1038/s41467-021-25184-4. PubMed PMID: 34385454; PubMed Central PMCID: PMCPMC8361191.

(13) Liu ZQ, Vazquez-Rodriguez B, Spreng RN, Bernhardt BC, Betzel RF, Misic B. Time-resolved structure-function coupling in brain networks. Commun Biol. 2022;5(1):532. Epub 2022/06/03. doi: 10.1038/s42003-022-03466-x. PubMed PMID: 35654886; PubMed Central PMCID: PMCPMC9163085.

(14) Zamani Esfahlani F, Faskowitz J, Slack J, Misic B, Betzel RF. Local structure-function relationships in human brain networks across the lifespan. Nat Commun. 2022;13(1):2053. Epub 2022/04/21. doi: 10.1038/s41467-022-29770-y. PubMed PMID: 35440659; PubMed Central PMCID: PMCPMC9018911.